



# Long-profile evolution of transport-limited gravel-bed rivers

**Andrew D. Wickert**[1] and **Taylor F. Schildgen**[2,3]

[1]Department of Earth Sciences and Saint Anthony Falls Laboratory, University of Minnesota,
Minneapolis, Minnesota, USA
[2]Institut für Erd- und Umweltwissenschaften, Universität Potsdam, 14476 Potsdam, Germany
[3]Helmholtz Zentrum Potsdam, GeoForschungsZentrum (GFZ) Potsdam, 14473 Potsdam, Germany

**Correspondence:** Andrew D. Wickert (awickert@umn.edu)

**Abstract.** Alluvial and transport-limited bedrock rivers constitute the majority of fluvial systems on Earth. Their long profiles hold clues to their present state and past evolution. We currently possess first-principles-based governing equations for flow, sediment transport, and channel morphodynamics in these systems, which we lack for detachment-limited bedrock rivers. Here we formally couple these equations for transport-limited gravel-bed river long-profile evolution. The result is a new predictive relationship whose functional form and parameters are grounded in theory and defined through experimental data. From this, we produce a power-law analytical solution and a finite-difference numerical solution to long-profile evolution. Steady-state channel concavity and steepness are diagnostic of external drivers: concavity decreases with increasing uplift, and steepness increases with an increasing sediment-to-water supply ratio. Constraining free parameters explains common observations of river form: to match observed channel concavities, gravel-sized sediments must weather and fine – typically rapidly – and valleys must widen gradually. To match the empirical square-root width–discharge scaling in equilibrium-width gravel-bed rivers, downstream fining must occur. The ability to assign a cause to such observations is the direct result of a deductive approach to developing equations for landscape evolution.

## 1 Introduction

Mountain and upland streams worldwide move clasts of gravel ( > 2 mm). Therefore, they consistently reshape their beds and – unless they are fully bedrock-confined – their bars and banks as well (Parker, 1978; Brasington et al., 2000, 2003; Church, 2006; Eke et al., 2014; Phillips and Jerolmack, 2016; Pfeiffer et al., 2017). Such rivers build and maintain topographic relief by carrying gravel out of the mountains. They can also transport sediment across moderate-relief continental surfaces and into sedimentary basins.

Geomorphologists commonly separate rivers into two broad categories based on the factor that limits their ability to change their long profile: detachment-limited and transport-limited (Whipple and Tucker, 2002). Detachment-limited rivers incise at a rate that is set by the mechanics of river incision into bedrock. Transport-limited rivers can incise or aggrade at a rate that is set by the divergence of sediment discharge through a river or valley cross section.

Here we present a new derivation for transport-limited gravel-bed river long-profile evolution that is based on relationships derived from theory, field work, and experimentation. We argue that developing this deductive approach – considering specific process relationships – is essential to advancing fluvial geomorphology and landscape evolution.

Much past work has focused on an inductive "stream-power" based formulation for detachment-limited river incision, in which the erosion rate is proportional to the drainage area (as a proxy for geomorphically effective discharge) and channel slope (e.g., Gilbert, 1877; Gilbert, 1877; Howard, 1980; Howard and Kerby, 1983; Whipple and Tucker, 1999; Gasparini and Brandon, 2011; Harel et al., 2016). This rule is intuitive, and may also be

described in terms of the rate of power dissipation against the river bed. However, such a generalized approach is agnostic to geomorphic processes. Efforts to understand the detailed mechanics of abrasion (Sklar and Dietrich, 1998, 2004; Johnson and Whipple, 2007) and quarrying (Dubinski and Wohl, 2013), the two main mechanisms of bedrock river erosion (Whipple et al., 2000), have aided efforts to generate mechanistic models for bedrock incision (Gasparini et al., 2006; Chatanantavet and Parker, 2009). However, the large number of measured parameters required for these relationships limits their use in practice and/or requires simplifications, such that the basic stream-power law remains the dominant model for detachment-limited rivers.

Writing a set of equations to describe the long-profile evolution of transport-limited gravel-bed rivers, in contrast, is aided by an extensive history of study that can be directly applied to models of long-profile evolution. This includes open-channel flow and flow resistance that can be applied to sediment-covered channels (e.g., Nikuradse, 1933; Keulegan, 1938; Limerinos, 1970; Aguirre-Pe and Fuentes, 1990; Parker, 1991; Clifford et al., 1992), bed-load transport (e.g., Shields, 1936; Meyer-Peter and Müller, 1948; Gomez and Church, 1989; Parker et al., 1998; Wilcock and Crowe, 2003; Wong and Parker, 2006; Bradley and Tucker, 2012; Furbish et al., 2012), and fluvial morphodynamics (e.g., Lane, 1955; Leopold and Maddock, 1953; Parker, 1978; Ikeda et al., 1988; Ashmore, 1991; Church, 2006; Pitlick et al., 2008; Eke et al., 2014; Bolla Pittaluga et al., 2014; Blom et al., 2016, 2017; Phillips and Jerolmack, 2016; Pfeiffer et al., 2017). Critical to the present work is the fact that the authors of these past studies have developed theory, tested it in both laboratory and field settings, and empirically determined the values of the relevant coefficients (e.g., Wong and Parker, 2006). Furthermore, bedrock channels can act as transport-limited systems (Johnson et al., 2009), meaning that an approach to transport-limited conditions may be able to describe the evolution of not only alluvial rivers, but rivers across much of Earth's upland surface. Based on this past research, we are able to write a simple and consistent set of equations for transport-limited gravel-bed river long-profile evolution that eschews tunable parameters, common in stream-power approaches to river long-profile evolution (Howard and Kerby, 1983; Whipple and Tucker, 1999, 2002) for those based on experimentation, measurements, and theory.

Here we link sediment transport and river morphodynamics to develop equations to describe gravel-bed river long profiles and, as a necessary extension, their tightly coupled width evolution. Our approach is complementary to a recent set of relations for alluvial river long profile shapes developed by Blom et al. (2016) and Blom et al. (2017), who explore equilibrium alluvial river long profile shapes in response to changes in grain size, slope, and width. Our approach and discussion are tailored to timescales from decades to millions of years, a broad range that results from the di-

rect derivation of these equations and their parameter values from fundamental physics, observations, and laboratory experiments. In particular, we (1) consider evolution of the full river valley, permitting analysis of timescales longer than those of channel filling; (2) follow Parker (1978) in allowing channel widths to self-form as a function of excess channel-forming shear stress; and (3) define channel roughness as a function of flow depth and grain size. Step (2) and (3) ultimately contribute to grain size canceling out of the final equation, leading to a relatively simple and applicable equation for gravel-bed river long-profile evolution in response to changes in water supply, sediment supply, and base level.

Our approach is outlined as follows: first, we generate fully coupled equations of gravel transport and fluvial morphodynamics to describe how channel long profiles change. Second, we investigate how the governing equations for gravel-bed rivers differ when we assume a channel with a self-formed equilibrium width vs. an externally set width. Third, we derive both analytical and numerical solutions for the case of an equilibrium-width channel, which is nearly ubiquitous in nature (Phillips and Jerolmack, 2016). Fourth, we quantify the constants for stream-power-based bed-load transport from Whipple and Tucker (2002) in a dimensionally consistent form that is based on our derived equations and the sizes of storm footprints. Fifth, we demonstrate that most gravel clasts in the landscape must be removed rapidly by weathering and/or downstream fining in order to produce rivers with concavities that lie within observed ranges. Sixth, we show that valley widening is required to produce rivers with observed concavities. Seventh, we investigate both steady-state and transient effects of base-level change (e.g., through tectonics) and the sediment-to-water discharge ratio (via climate and/or tectonics) on river long profiles, and demonstrate that the former changes concavity while the latter changes steepness. Eighth and finally, we derive that downstream fining and channel concavity must combine to be the mechanistic cause of channel width scaling with the square root of water discharge ($b \propto Q^{0.5}$) (Lacey, 1930; Leopold and Maddock, 1953), at least in equilibrium-width (including near-threshold) transport-limited gravel-bed rivers.

## 2   Derivations

We consider gravel-bed rivers to exist in one of two states: equilibrium-width and fixed-width. In the first, we assume that the channel-forming (i.e., bank-full) shear stress on the bed remains a constant ratio of the critical shear stress that sets the threshold for initiation of sediment motion (after Parker, 1978). The channel width is set to maintain this ratio. In the second, the channel and valley width are assumed to be identical in order to use the one-dimensional form of the sediment continuity equation, called the Exner equation (e.g., Paola et al., 1992; Whipple and Tucker, 2002; Blom

et al., 2016). A third and more general case exists in which one externally imposes both channel and valley width. We do not address this case here, although it may be solved using the equations provided.

Our primary focus here is on equilibrium-width rivers, which are common throughout the world (Phillips and Jerolmack, 2016; Pfeiffer et al., 2017). Most maintain a bed shear stress that is slightly greater than that for the initiation of motion (Parker, 1978; Phillips and Jerolmack, 2016), and this near-threshold condition is characteristic of both fully alluvial and alluvial-mantled bedrock streams (Phillips and Jerolmack, 2016). Rivers in rapidly uplifting mountain belts maintain a bed shear stress that can be much greater than that for the initiation of particle motion; this results in higher sediment discharges that help to balance the high inputs of sediment that result from rock uplift (Pfeiffer et al., 2017). Although these rivers do not exist in a near-threshold state, they maintain an equilibrium width corresponding to their ratio of bed shear stress to critical shear stress for the initiation of motion that allows them to transport the sediment that they are supplied (Pfeiffer et al., 2017).

We split our derivations into sections on equilibrium-width (Sect. 2.1) and fixed-width (Sect. 2.2) rivers. We first develop a sediment-discharge relationship as a function of channel morphology. This portion of the derivation can apply to both alluvial (transport-limited) and bedrock (both transport- and detachment-limited) rivers. Simulating detachment-limited rivers in which abrasion is the dominant mechanism of river incision requires sediment-flux-dependent erosion relationships (Sklar and Dietrich, 2001; Whipple and Tucker, 2002; Sklar and Dietrich, 2004; Gasparini et al., 2006, 2007), which we do not discuss in detail here. We focus on alluvial and transport-limited bedrock cases by applying a statement of sediment volume balance (the Exner equation) to develop a differential equation that describes alluvial river long-profile evolution over time. The width closure for the equilibrium-width gravel-bed river produces a mathematically clean solution from which intuition can be readily gained, and this is the focus of our discussion. The fixed-width case, which is characteristic of an engineered gravel-bed river with rigid walls, is included for contrast with the equilibrium-width case and comparison with studies in which an externally set width is assumed (e.g., Blom et al., 2016, 2017).

## 2.1 Equilibrium-width river

We derive an equation for the evolution of the long profile of an equilibrium-width gravel-bed river that lies within a valley whose shape is arbitrary (although at least as wide as the channel) and may evolve through time. We first state a modified Exner equation for the conservation of bed-load sediment discharge ($Q_s$) for a river in a valley of width $B$ (Fig. 1):

$$\frac{\partial z}{\partial t} = -\frac{1}{1-\lambda_p}\left(\frac{1}{B}\frac{\partial Q_s}{\partial x} - \frac{Q_s}{B^2}\frac{\partial B}{\partial x}\right). \tag{1}$$

Here, $z$ is the elevation of the river bed surface, and is often also denoted as $\eta$ in the alluvial river literature. Time is represented by $t$. $\lambda_p$ is porosity, for which 0.65 is a representative value (consistent with Beard and Weyl, 1973). $x$ is down-valley distance, which is the same as down-channel distance only for a straight river flowing directly down-valley. $B$ is the width of the river valley at the current level of the river bed; it may change with changes in river bed elevation and/or as the valley widens or narrows over time. These and all variables are defined in Appendix A. $\lambda_p$ and $B$ scale the result: a higher porosity means that less sediment must be eroded or deposited to produce the same change in bed elevation (i.e., aggradation or incision). A wider valley means that more sediment must be moved to produce a given amount of aggradation or incision.

Equation (1) differs from the original form that Exner (1920, 1925) developed (Eq. B1), which considers only channel-width-averaged sediment discharge (e.g., Paola et al., 1992; Paola and Voller, 2005). This is appropriate for aggradation or incision within a channel or in a vertically walled valley that is exactly one channel width wide, but is unable to be solved for aggradation or incision for the common case of a valley that is wider than the channel. Because the evolving landform is the valley, we have chosen $x$ to be down-valley distance, and describe the steps required to link channel-scale dynamics to longer-term long-profile evolution using our modified Exner equation (Eq. 1) for sediment continuity in Appendix B1.

Following this definition of a sediment continuity equation, we take several steps towards developing a simple formulation for the total discharge of sediment through the river, $Q_s$. Once we find the correct expression for this value, we insert it into Eq. (1), which we then simplify into a final differential equation for transport-limited gravel-bed river long-profile evolution.

Towards this eventual goal, our second step is to define bed-load sediment discharge per unit width, $q_s$, where

$$q_s = \frac{Q_s}{b}. \tag{2}$$

Here, $b$ is the width (breadth) of the river channel ($b \leq B$). We compute bed-load transport using the Wong and Parker (2006) formulation of the Meyer-Peter and Müller (1948) formula. This formula is semi-empirical: its core form is based on a balance of shear stress along the bed driving particle motion and particle weight resisting that motion, but its power-law functional form as well as its coefficients and exponents are fit to the results of laboratory experiments. More fully theory-based formulations are under development (Furbish et al., 2012; Fathel et al., 2015) and promise signifi-

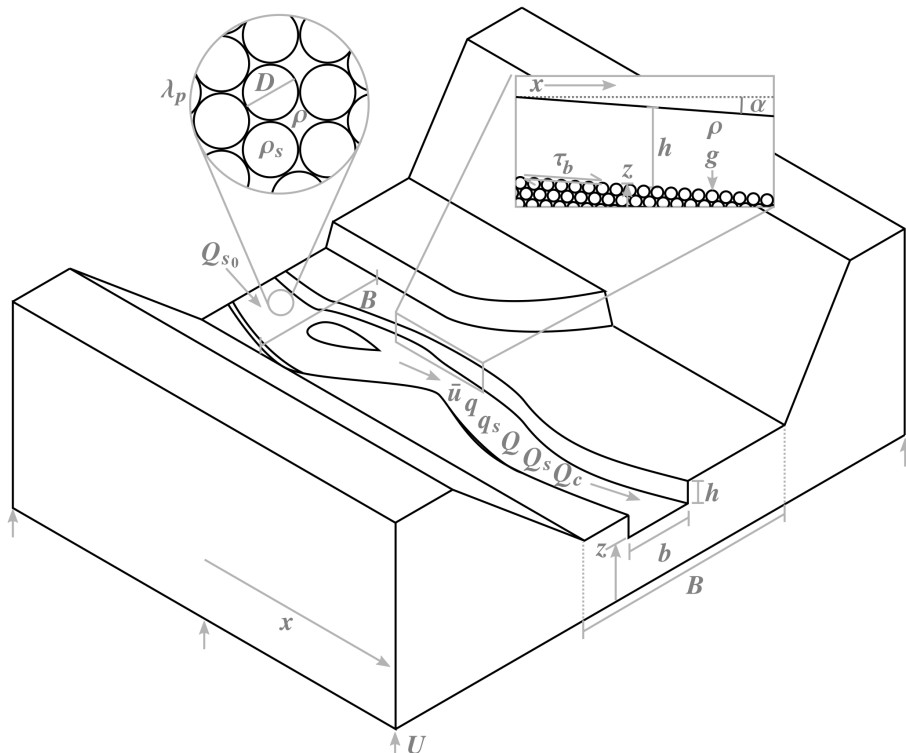

**Figure 1.** Schematic block diagram of sediment transport through a reach of a transport-limited river. Variables are defined in the text and in Appendix A. The balance of sediment input, sediment output, and uplift determine whether the river bed at each point downstream will rise, fall, or remain at a constant elevation.

cant advances in our understanding and prediction of sediment transport. Our choice to use the Meyer-Peter and Müller (1948) formulation stems from its longevity, its simplicity, the fact that it has been well tested (Wong and Parker, 2006), and its compatibility with the channel-width closure resulting from the work of Parker (1978). We stress that our general set of steps to deriving equations for long-profile evolution may be repeated for any sediment-transport relation.

$$q_{\mathrm{s}} =$$
$$\begin{cases} 0 & \text{if } \tau_{\mathrm{b}}^* \leq \tau_{\mathrm{c}}^* \\ \phi\left(\frac{\rho_{\mathrm{s}}-\rho}{\rho}\right)^{1/2} g^{1/2} \left(\tau_{\mathrm{b}}^* - \tau_{\mathrm{c}}^*\right)^{3/2} D^{3/2} & \text{if } \tau_{\mathrm{b}}^* > \tau_{\mathrm{c}}^*. \end{cases} \quad (3)$$

Here, $\phi = 3.97$ (Wong and Parker, 2006) is an experimentally derived sediment-transport rate coefficient. $\rho_{\mathrm{s}}$ is sediment density, $\rho$ is water density, and $g$ is acceleration due to gravity. $\left|\tau_{\mathrm{b}}^*\right|$ is the magnitude of the dimensionless basal shear stress (defined in Eq. 6, below), and is also called the "Shields stress" (after Shields, 1936). $\tau_{\mathrm{c}}^* = 0.0495$ (Wong and Parker, 2006) is the experimentally derived dimensionless critical shear stress for the initiation of particle motion, and is also called the "critical Shields stress". $D$ is a representative sediment grain (particle) size, which we take to be the median gravel clast diameter. This formula is technically for sediment-transport capacity, $Q_{\mathrm{c}}$, per unit channel width,

but in a transport-limited river, sediment is always supplied at or above capacity such that $Q_{\mathrm{s}} \equiv Q_{\mathrm{c}}$. We assume that we know the downstream direction; a supplement to this derivation in which we explicitly consider directionality is included in Appendix C in order to streamline the main text.

While the Meyer-Peter and Müller (1948) equation is only strictly valid for a single grain size class, it is often an acceptable approximation for natural rivers with multiple size classes (Gomez and Church, 1989; Paola and Mohrig, 1996). Interactions among multiple grain size classes may cause a condition of "equal mobility" in gravel-bed rivers (e.g., Parker et al., 1982): small grains become trapped inside pits between larger grains, while large grains rest on a carpet of smaller grains and thus are exposed to more of the force of the flow. Even where significant deviations from equal mobility are observed, $\tau_{\mathrm{c}}^*$ for the 50th percentile grain size ($D_{50}$) remains constant (Komar, 1987; Komar and Shih, 1992). For the representative grain size ($D$) in Eq. (3) (and Eq. 27, below), Wong and Parker (2006) used the mean size of uniform gravel. We suggest the median grain size ($D_{50}$) as representative of $D$ for the mixed-size sediment of natural rivers due to its relative ease of standardized measurement (Wolman, 1954) and constant dimensionless critical shear stress for the initiation of motion (Komar, 1987; Komar and Shih, 1992; Paola and Mohrig, 1996). Regardless of this choice, $D$ can-

cels out in our formulation for equilibrium-width gravel-bed rivers, starting in Eq. (18).

Basal shear stress induces a drag force on the grains and drives sediment transport. To compute this basal shear stress ($\tau_b$), we invoke the normal flow (steady, uniform) assumption, the wide-channel approximation ($b \gg h$, where $h$ is the flow depth), and the small-angle formula (Fig. 1, upper right inset):

$$\tau_b = \rho g h \sin \alpha \tag{4}$$
$$\approx \rho g h S.$$

Here, $\alpha$ is the angle between the plane of the water surface and the horizontal in the downstream direction, and $S$ is the channel slope. The water surface and bed surface slopes are assumed to be parallel (following the normal flow assumption). Assuming that the flow is from left to right, we can define channel slope as

$$S = -\frac{1}{\mathbb{S}}\frac{dz}{dx}. \tag{5}$$

The above equation includes the sinuosity (river length divided by valley length, $\mathbb{S}$) of the channel in the valley; this is necessary to convert the channel slope, which drives sediment transport, into the valley slope, which follows the $x$ coordinate orientation used in Eq. (1) (see Appendix B). The negative sign is used to denote direction, but is included for convenience and intuition rather than for mathematical precision. When slope is raised to a power, only the magnitude of the slope is affected, with the "$-$" sign being applied afterwards.

The drag force on sediment grains induced by basal shear stress is resisted by the submerged weight of the grains. The ratio of these forces defines the Shields stress:

$$\tau_b^* = \frac{\tau_b D^2}{(\rho_s - \rho)g D^3} = \frac{\tau_b}{(\rho_s - \rho)g D}. \tag{6}$$

In gravel-bed rivers, all of the shear stress is assumed to act as skin friction, meaning that it is directly imparted to the particles instead of being partially absorbed as form drag on larger-scale features (e.g., bedforms). When this dimensionless stress is in excess of the critical Shields stress ($\tau_c^*$), particles begin to move.

In equilibrium-width gravel-bed rivers, the dimensionless basal shear stress at the channel-forming discharge is assumed to be maintained as a constant multiple of the dimensionless critical shear stress for initiation of sediment motion (Parker, 1978). This proportionality may be equally represented by dimensional stresses; we use the dimensionless Shields stresses here for consistency:

$$\tau_b^* = (1 + \epsilon)\tau_c^*. \tag{7}$$

Parker (1978) derived that $\epsilon \approx 0.2$ for self-formed gravel-bed rivers with mobile banks made of the same size gravel as the

bed, based on theory and channel geometry. This value has been found empirically and near-universally in rivers around the world outside of rapidly tectonically uplifting environments (Phillips and Jerolmack, 2016; Pfeiffer et al., 2017). $(1 + \epsilon)\tau_c^*$ is the dimensionless shear stress experienced by the bed of the channel when the shear stress experienced by the banks is equal to $\tau_c^*$. The Parker (1978) near-threshold gravel-bed river solution states that any excess stress would cause the banks to erode and the channel to widen, reducing the flow depth, and thereby decreasing $\tau_b^*$ to $(1 + \epsilon)\tau_c^*$.

The channel-forming discharge, also termed the geomorphically effective discharge, is equivalent to the bank-full flow in a self-formed gravel-bed river with gravel bars and banks. Blom et al. (2017) derived a method to differentiate the channel-forming discharge, defined as that required to maintain the channel slope, from the most effective discharges to move different grain size classes of sediments. This is a significant distinction, but one that will not be necessary for our modeling approach, as we consider only the discharges that are large enough to cause non-negligible geomorphic change. In a self-formed gravel-bed river, a near-threshold state is maintained in which $\tau_b^* = 1.2\tau_c^*$ (Parker, 1978). We use this ratio between applied and critical shear stress to compute the numerical values for constants given in this derivation.

Substituting $\tau_b^*$ in Eq. (3) with its value given in Eq. (7) reduces the complexity of Eq. (3) by converting its excess shear stress terms at a channel-forming discharge ($\tau_b^* - \tau_c^*$) into a constant (by a factor of $\epsilon$) and requiring that only the case with a positive nonzero $q_s$ be a plausible solution:

$$q_s = \phi\left(\frac{\rho_s - \rho}{\rho}\right)^{1/2} g^{1/2} \epsilon^{3/2} \tau_c^{*3/2} D^{3/2} \tag{8}$$
$$= k_{q_s} D^{3/2}.$$

In an equilibrium-width gravel-bed river, $q_s$ is a function of grain size alone. The value of $k_{q_s} = 0.0157$ is obtained from $\phi = 3.97$ (Wong and Parker, 2006), $\rho_s = 2650\,\mathrm{kg\,m^{-3}}$ (density of quartz), $\rho = 1000\,\mathrm{kg\,m^{-3}}$ (density of water), $g = 9.807\,\mathrm{m\,s^{-2}}$, $\epsilon = 0.2$ (for a threshold-width channel; Parker, 1978), and $\tau_c^* = 0.0495$ (Wong and Parker, 2006).

It may be counterintuitive that sediment discharge per unit width increases with grain size. This is a result of the equilibrium-width argument. Channel geometry adjusts to maintain a constant excess basal shear stress regardless of grain size. However, larger grains have a greater vertical dimension: many small grains rolling or sliding along the bed will displace less mass than a single larger grain.

Equation (8) is physically valid only where $b > D$ (see Eq. 16, below) and is a good approximation only where $b \gg h$ and $h > D$ (see Eq. 9, below). It seems likely that, at a flow width that is some small multiple of $D$, an equilibrium-width gravel-bed channel would be replaced by a boulder cascade or similar system that is more dispersed. While we do not investigate the exact point of this process-domain

boundary, this forms a practical limit to the theory presented here.

For a self-formed gravel-bed channel, channel depth must satisfy Eq. (7). Using the normal flow assumption, the depth–slope product (Eq. 4) defines basal shear stress. Inserting the dimensionless basal shear stress (calculated by combining Eqs. 4 and 6) into Eq. (7) and rearranging to solve for $h$ at a channel-forming discharge results in

$$h = \frac{\rho_s - \rho}{\rho}(1 + \epsilon)\tau_c^* \frac{D}{S}. \qquad (9)$$

Next, we compute mean water flow velocity ($\overline{u}$) for a geomorphically effective flow. We solve for mean flow velocity using the empirically derived Manning–Strickler formulation (following Parker, 1991) of the Chézy equation. We first write the Chézy equation for steady, uniform flow,

$$\overline{u} = C_z \sqrt{ghS}. \qquad (10)$$

Here, $C_z$ is a factor that relates flow velocity to shear velocity, and $\sqrt{ghS}$ is the shear velocity for steady, uniform flow. We then define $C_z$, following the Manning–Strickler formulation, as

$$C_z = 8.1 \left(\frac{h}{\lambda_r}\right)^{1/6}. \qquad (11)$$

The coefficient of 8.1 is empirical (Parker, 1991). $\lambda_r$ is the characteristic roughness length scale; this is often denoted as $k_s$, but we reserve this notation for the channel steepness index in slope–area space (Sect. 5.2). The flow depth ($h$) in the numerator and the roughness ($\lambda_r$) in the denominator indicate that flow velocity increases with distance to the no-slip boundary, and decreases with increasing boundary roughness. The gravel clasts themselves are the major source of roughness (and therefore flow resistance) in a gravel-bed river. Clifford et al. (1992) related grain size to roughness length to obtain the approximation that $\lambda_r \approx 6.8D$, where $D$ is the median gravel clast diameter. Carrying this forward, but using a standard "equals" sign, produces an expression for flow velocity that depends only on constants and basic geomorphic parameters:

$$\overline{u} = 5.9g^{1/2}\frac{h^{2/3}S^{1/2}}{D^{1/6}}. \qquad (12)$$

The power-law form of the empirically developed Manning–Strickler formulation (see Parker, 1991) closely approximates the more theoretical logarithmic boundary layer approach of Keulegan (1938) for ratios of depth to roughness length that are characteristic of gravel-bed rivers; thus, the former is an equally accurate and more mathematically convenient approach.

Water discharge per unit width can be computed by multiplying $\overline{u}$ by $h$ as follows:

$$q = \overline{u}h = 5.9g^{1/2}\frac{h^{5/3}S^{1/2}}{D^{1/6}}. \qquad (13)$$

Substituting Eq. (9) into Eq. (13) gives

$$q = \overline{u}h = 5.9g^{1/2}\left(\frac{\rho_s - \rho}{\rho}\right)^{5/3}(1 + \epsilon)^{5/3}\tau_c^{*5/3}\frac{D^{3/2}}{S^{7/6}}. \qquad (14)$$

The final equation that we require to obtain channel width ($b$) for Eq. (2) is that for continuity. We approximate the channel cross section as rectangular such that the magnitude of the channel-forming water discharge, $Q$, is equal to the product of the flow speed, width, and depth:

$$Q = \overline{u}bh = qb. \qquad (15)$$

Rearranging Eq. (15) to solve for $b$, and then substituting Eq. (14) for $q$, yields

$$b = 0.17g^{-1/2}\left(\frac{\rho_s - \rho}{\rho}\right)^{-5/3}(1 + \epsilon)^{-5/3}\tau_c^{*-5/3}\frac{QS^{7/6}}{D^{3/2}} \qquad (16)$$

$$= k_b \frac{QS^{7/6}}{D^{3/2}}.$$

Equation (16) predicts the equilibrium width of a river channel that has a constant ratio of basal Shields stress to critical Shields stress (Eq. 7), following Parker (1978). This equilibrium width is set by a trade-off between discharge and slope, both increasing basal Shields stress, and grain size, which decreases the basal Shields stress. To focus attention on these key variables ($Q$, $S$, and $D$, respectively), we lump the constants into $k_b = 2.61$, assuming $\epsilon = 0.2$ (Parker, 1978; Phillips and Jerolmack, 2016).

Finally, channel width ($b$) and sediment discharge per unit width ($q_s$, Eq. 8) can be multiplied together to yield $Q_s$. In order to relate this product to the field, we include an additional term, the intermittency ($I$), which is the fraction of the total time that a river produces a geomorphically effective flow (after Paola et al., 1992); smaller flows are considered to be unable to produce non-negligible geomorphic change. For example, if the annual flood on a self-formed gravel-bed river is a bank-full event, and this event lasts for 3–4 days, $I \approx 0.01$; such conditions are typical for rainfall-fed midlatitude rivers.

We express this equation first in terms of magnitudes,

$$Q_s = k_{Q_s}IQS^{7/6}. \qquad (17)$$

We then return directionality to the equation by replacing $S$ following Eq. (5), and noting that the sign is applied after raising its argument to a power. (See Appendix C for a very brief discussion of the use of slope, $S$, in place of separate terms for its direction and magnitude.)

$$Q_s = -\frac{k_{Q_s}I}{\mathbb{S}^{7/6}}Q\frac{dz}{dx}\left|\frac{dz}{dx}\right|^{1/6}. \qquad (18)$$

In both of these equations,

$$k_{Q_s} = k_{q_s} k_b \qquad (19)$$

$$= \frac{0.17 \phi \epsilon^{3/2}}{\left(\frac{\rho_s - \rho}{\rho}\right)^{7/6} (1+\epsilon)^{5/3} \tau_c^{*1/6}}$$

$$= 0.041.$$

The numerical value for $k_{Q_s}$ is provided for $\epsilon = 0.2$, following Parker (1978). While we treat $\epsilon$ as a constant here, recent research by Pfeiffer et al. (2017) indicates that its value may vary. Furthermore, it is important to note that by using a rectangular channel assumption, we neglect the potentially important component of spatial variability in flow depth. This variability, which is especially common in braided systems, can result in deep scours that increase the net bed-load sediment-transport capacity of the river channel (Paola et al., 1999). This unaccounted for variability may, therefore, significantly increase $k_{Q_s}$ beyond what is predicted here.

Equation (18) demonstrates that in an equilibrium-width river, sediment discharge obeys a stream-power relationship (Paola et al., 1992; Whipple and Tucker, 2002) in which the values of the coefficient and exponents are defined based on the above derivation. Although it is beyond the scope of this work on transport-limited rivers, the derivation of transport capacity to this point may be useful for studies of sediment-flux-dependent detachment-limited river incision (Gasparini et al., 2006, 2007; Hobley et al., 2011).

Hydraulic geometry adjustment in an equilibrium-width gravel-bed river causes bed-load sediment discharge to be independent of grain size. Sediment discharge per unit width increases with grain size as $q_s \propto D^{3/2}$ (Eq. 8). Channel width, in comparison, decreases as grain size increases, $b \propto D^{-3/2}$ (Eq. 16), due to the scaling relationships between grain size and both channel depth and flow resistance (Eqs. 9 and 14).

In this derivation, we hold $\tau_c^*$ constant instead of making it a function of slope to the 1/4 power, as has been suggested by Lamb et al. (2008) based on experimental and field data. We do so for three reasons. First, a constant critical Shields stress is appropriate for rivers with slopes that are $\lesssim 0.03$ (Lamb et al., 2008); this set comprises most rivers in the world. Second, the assumption of an equilibrium-width river (Parker, 1978) results in the removal of the threshold associated with $\tau_c^*$ from the sediment-transport equation. Third, the remaining slope dependence is to the 1/24 power (Eq. 19). Adding such a weak slope dependence that may marginally improve accuracy would introduce a mathematically significant nonlinearity into the system of equations, thereby impeding our goal of providing intuition into the behavior of gravel-bed rivers.

While $q$ and $q_s$ are defined in the down-channel direction, $Q$ and $Q_s$ are equal for both the down-channel and down-valley directions. This convenient equality results geometrically from the fact that, as the angle between a river center-line and a line that crosses the valley perpendicularly increases, the flux (width-normalized discharge) decreases, but the fraction of the line occupied by river increases (Fig. B2). This decrease and increase are proportional, and thus cancel one another out. One may also consider this to be the result of path independence: the discharge that exits a segment of valley must be equal to the discharge that enters it (Appendix B2).

We combine Eqs. (1) and (18) with a source/sink term for uplift (or subsidence) to produce a long-profile evolution equation for a transport-limited gravel-bed river:

$$\frac{\partial z}{\partial t} = \frac{k_{Q_s} I}{\mathbb{S}^{7/6} (1 - \lambda_p)} \left[ \frac{7}{6} \frac{1}{\left(\frac{\partial z}{\partial x}\right)} \frac{\partial^2 z}{\partial x^2} + \frac{1}{Q} \frac{\partial Q}{\partial x} - \frac{1}{B} \frac{\partial B}{\partial x} \right] \qquad (20)$$
$$\frac{Q}{B} \frac{\partial z}{\partial x} \left| \frac{dz}{dx} \right|^{1/6} + U.$$

This equation has the general form of a nonlinear diffusion equation, with the nonlinearity being a combination of $|dz/dx|^{1/6}$ and any possible nonlinear relationships that arise in $Q(x)$ and $B(x)$. To the right of the equals sign, the leftmost term is a collection of constants. The brackets hold the gradients in slope, water discharge, and valley width. To the right of the brackets are the main drivers: long-profile response rates increase with increasing discharge magnitude and slope, both of which speed sediment transport, and response rates decrease as valley width increases, which creates more space that must be filled or emptied to produce a change in river-bed elevation. By placing sinuosity with the constants, we assume that it changes in space only gradually, if at all. This equation would simplify to the linear diffusional relationship derived by Paola et al. (1992) if we (1) considered a constant bed roughness instead of including the Manning–Strickler-based flow resistance that introduces a depth dependence (Eq. 12), (2) removed the effects of variable valley width, and (3) considered a uniform water discharge.

Uplift and subsidence ($U$) are not the only possible source and sink for material: Murphy et al. (2016) note the importance of chemical weathering, which must remove mass from rock, and Shobe et al. (2016) investigate the importance of local colluvial input to rivers. We do not focus on either of these here, but note that the latter must also be related to valley width evolution, which may produce enhanced hillslope sediment inputs, for example, through bank collapse and landsliding.

Equation (20) describes the long-profile evolution of an equilibrium-width gravel-bed alluvial river. The dependencies of the variables in Eq. (20) are as follows:

$$z = z(x, t) \qquad (21)$$
$$Q = Q(x, t) \qquad (22)$$
$$B = B(z(x, t), t) \qquad (23)$$
$$U = U(x, t). \qquad (24)$$

The dependency of valley width, $B$, on the elevation of the river bed, $z$, is the result of the fact that few valleys have vertical walls. Therefore, changes in valley elevation produce changes in valley width, even in absence of time-evolution of the valley geometry that then feeds back into the rate of long-profile evolution. Mathematically, this adds an arbitrary dependence on $z$ that limits the analytically solvable forms of Eq. (20).

## 2.2 Fixed-width river

If the width of the river is externally known and is identical to the width of the valley, another solution is possible. To produce this solution, we first simplify the Exner equation to its one-dimensional form for the case in which $b = k_{b,B}B$, in which the constant coefficient $k_{b,B} \leq 1$. By expanding $Q_s = q_s b$ and canceling out width:

$$\frac{\partial z}{\partial t} = -\frac{k_{b,B}}{1 - \lambda_p}\frac{\partial q_s}{\partial x}. \tag{25}$$

Combining this form of the Exner equation with the Wong and Parker (2006) version of the Meyer-Peter and Müller (1948) gravel transport formula, given in Eq. (3), and assuming that $\tau_b^* \geq \tau_c^*$, leads to the following differential equation for gravel-bed river long-profile evolution:

$$\frac{\partial z}{\partial t} = \frac{k_{b,B}}{1 - \lambda_p}\frac{3}{2}\phi\left(\frac{\rho_s - \rho}{\rho}\right)^{1/2} g^{1/2}\left(\tau_b^* - \tau_c^*\right)^{1/2} D^{1/2} \tag{26}$$
$$\left[D\frac{\partial \tau_b^*}{\partial x} + \left(\tau_b^* - \tau_c^*\right)\frac{\partial D}{\partial x}\right].$$

Here, no form of width closure is assumed. We maintain the assumption that $\tau_c^*$ is a constant, meaning that this equation is valid for rivers of slopes that are $\lesssim 0.03$ (Lamb et al., 2008). This simplification is included both for comparison with Eq. (20) for equilibrium-width rivers and to avoid the added mathematical complexity of including a weak nonlinearity.

Equation (26) hides discharge, width, slope, and an additional grain-size dependence within $\tau_b^*$. To include these explicitly, we combine Eq. (15) and (12) to solve for flow depth, $h$, and insert this depth into the Meyer-Peter and Müller (1948) sediment-transport formula (Eq. 3) via the definition of dimensionless basal shear stress given in Eq. (6):

$$q_s = \tag{27}$$
$$\begin{cases} 0 & \text{if } |\tau_b^*| \leq \tau_c^* \\ -\text{sgn}\left(\frac{dz}{dx}\right)\phi\left(\frac{\rho_s - \rho}{\rho}\right)^{1/2} g^{1/2}\left(\frac{0.345}{g^{3/10}\mathbb{S}^{7/10}}\right) \\ \frac{1}{\frac{\rho_s - \rho}{\rho}}\frac{1}{D^{9/10}}\left(\frac{Q}{b}\right)^{3/5}\left|\frac{\partial z}{\partial x}\right|^{7/10} - \tau_c^*\right)^{3/2} D^{3/2} & \text{if } |\tau_b^*| > \tau_c^*. \end{cases}$$

Here, the signum function and absolute values are included to allow for flow and sediment transport in either the positive

or negative $x$ direction (see Appendix C). In a natural river, $q_s$ is combined with an intermittency, $I$, which is equal to the fraction of the time that the discharge is geomorphically effective; smaller discharges are assumed to carry negligible bed-load sediment (Paola et al., 1992).

To formulate the differential equation for long-profile evolution of a transport-limited gravel-bed river of arbitrary width, we combine our transport relationship (Eq. 27) with our statement of volume balance (Eq. 25). In the following equation, we again consider only flows in which $\tau_b^* > \tau_c^*$; to use it in practice, one would first run a check as to whether $\tau_b^* > \tau_c^*$. If true, the bed would evolve as shown; if false, $\partial z/\partial t = 0$.

$$\frac{\partial z}{\partial t} = \frac{3}{2}\frac{k_{b,B}\phi g^{1/2}I}{1 - \lambda_p}\left(\frac{\rho_s - \rho}{\rho}\right)^{1/2} \tag{28}$$
$$\left(\frac{1}{\frac{\rho_s - \rho}{\rho}}\frac{0.345}{g^{3/10}\mathbb{S}^{7/10}}\left|\frac{\partial z}{\partial x}\right|^{7/10}\frac{1}{D^{9/10}}\frac{Q^{3/5}}{b^{3/5}} - \tau_c^*\right)^{1/2} D^{1/2}$$
$$\left[\frac{Q^{3/5}D^{1/10}}{b^{3/5}}\left|\frac{\partial z}{\partial x}\right|^{7/10}\left(\frac{3}{5}\frac{1}{Q}\frac{\partial Q}{\partial x} - \frac{3}{5}\frac{1}{b}\frac{\partial b}{\partial x}\right.\right.$$
$$\left.+ \frac{7}{10}\frac{1}{\left|\frac{\partial z}{\partial x}\right|}\frac{\partial^2 z}{\partial x^2} - \frac{9}{10}\frac{1}{D}\frac{\partial D}{\partial x}\right) + \left(\frac{1}{\frac{\rho_s - \rho}{\rho}}\right.$$
$$\left.\frac{0.345}{g^{3/10}\mathbb{S}^{7/10}}\left|\frac{\partial z}{\partial x}\right|^{7/10}\frac{1}{D^{9/10}}\frac{Q^{3/5}}{b^{3/5}} - \tau_c^*\right)\frac{\partial D}{\partial x}\right] + U.$$

When $b$ is set such that Eq. (7) for an equilibrium-width gravel-bed channel holds true and $b = B$, Eq. (28) becomes equal to Eq. (20).

In addition to the variable space–time dependencies listed in Eqs. (21)–(24), we include the following two for Eq. (28):

$$b = b(z(x, t), t) = B(z(x, t), t) \tag{29}$$
$$D = D(x, t). \tag{30}$$

## 3 Analytical solutions

Two analytical solutions are presented here to help build intuition into the shape of gravel-bed river long profiles. The most generally applicable of these, for an equilibrium-width gravel-bed river that is neither aggrading nor incising in an area with no tectonic activity, is presented first. This solution is a power law that relates measurable hydrologic and landscape parameters to river long-profile shape. The second analytical solution is for a fixed-width river that adds the additional assumptions that width, discharge, and grain size are held constant. This solution provides an equilibrium transport slope.

### 3.1 Relationships between width, discharge, drainage area, and downstream distance

In order to analytically solve special cases of the provided equations for river channel long-profile evolution, we need a way to write Eq. (20) in terms of only $z$ and $x$, meaning that we should rewrite $Q$ and $B$ in terms of $x$. For any real river, there is a measurable relationship between discharge and distance downstream. Such relationships, and others in this paper, have a power-law form. In order to write these in a consistent and intuitive way, all power-law coefficients are designated $k$ and all exponents ("powers") are designated $P$. Each coefficient and exponent is given a two-letter subscript where the first letter indicates the variable from which one is converting (right-hand side) and the second letter indicates the variable to which one is converting (left-hand side).

Based on observations (Hack, 1957; Costa and O'Connor, 1995)

$$Q = k_{A,Q} A^{P_{A,Q}} \tag{31}$$

$$A = k_{x,A} x^{P_{x,A}}. \tag{32}$$

$Q$ in Eq. (31) refers to the discharge of a geomorphically effective flood – in our case, this is one that applies a shear stress $\tau_b^* \approx (1+\epsilon)\tau_c^*$ (Wolman and Miller, 1960; Parker, 1978; Sullivan and Lucas, 2007). $P_{x,A} \approx 4/7$ in the inverse of the Hack exponent (Gray, 1961; Maritan et al., 1996; Birnir et al., 2001). Substituting $A$ in Eq. (31) with Eq. (32) provides the needed transfer function between $Q$ and $x$:

$$Q = k_{A,Q} k_{x,A}^{P_{A,Q}} x^{P_{x,A} P_{A,Q}} \tag{33}$$

$$= k_{x,Q} x^{P_{x,Q}}. \tag{34}$$

These equations are continuum idealizations of a river with a tributary network. Real rivers experience discrete jumps in water discharge at tributary junctions. The smooth curves of water discharge vs. down-valley distance produced by these relationships, in contrast, are beneficial for building intuition.

Solutions to Eq. (20) also depend on how valley width, $B$, changes with distance downstream. Following Snyder et al. (2000) and Tomkin et al. (2003), who formulated a power-law relationship between valley width and drainage area, we propose that $B$ is also a power-law function of $x$:

$$B = k_{x,B} x^{P_{x,B}}. \tag{35}$$

### 3.2 Equilibrium-width river

In order to develop an analytical solution to Eq. (20), we first replace $Q$ and $B$ using Eqs. (31)–(35):

$$\frac{\partial z}{\partial t} = \frac{k_{Q_s} I}{\mathbb{S}^{7/6} (1 - \lambda_p)} \left[ \frac{7}{6} \frac{1}{\left(\frac{\partial z}{\partial x}\right)} \frac{\partial^2 z}{\partial x^2} + \frac{P_{x,Q}}{x} - \frac{P_{x,B}}{x} \right] \tag{36}$$

$$\frac{k_{x,Q} x^{P_{x,Q}}}{k_{x,B} x^{P_{x,B}}} \left( \frac{\partial z}{\partial x} \right) \left| \frac{\partial z}{\partial x} \right|^{1/6} + U.$$

One useful analytical solution to this equation would be that for the steady-state case, in which

$$\frac{\partial z}{\partial t} = 0. \tag{37}$$

However, no analytical solution exists for this form of the equation when tectonic uplift or subsidence is present. As a close substitute, and one that can greatly simplify Eq. (36), we choose the case in which the river is neither aggrading nor incising. Therefore, its only vertical motion is as it passively rides up or down on the Earth's surface:

$$\frac{\partial z}{\partial t} = U. \tag{38}$$

For the special case in which there is no uplift, Eq. (37) holds. It is important to note that this case implies a continuous externally sourced sediment supply in order to maintain a fixed topography without relative uplift across the stream profile.

For such a no-uplift steady-state condition to persist over geologic time requires a constant input of sediment from the hillslopes. This may be reasonable for a river that reaches an equilibrium long profile much more rapidly than the surrounding landscape evolves and its relief changes. It is also useful as a benchmark for numerical solutions (Fig. 2).

Applying Eq. (38) to Eq. (36) yields a second-order nonlinear ordinary differential equation that is analytically solvable:

$$0 = \frac{7}{6} \frac{1}{\left(\frac{dz}{dx}\right)} \frac{d^2 z}{dx^2} + \frac{P_{x,Q} - P_{x,B}}{x}. \tag{39}$$

Its solution is a power law, solved using two known points along the long profile – $(x_0, z_0)$ and $(x_1, z_1)$. Practical choices for these points are the upstream and downstream boundaries of the river segment being studied:

$$z = \tag{40}$$

$$(z_1 - z_0) \left( \frac{x^{(1+6(P_{x,B}-P_{x,Q})/7)} - x_0^{(1+6(P_{x,B}-P_{x,Q})/7)}}{x_1^{(1+6(P_{x,B}-P_{x,Q})/7)} - x_0^{(1+6(P_{x,B}-P_{x,Q})/7)}} \right) + z_0.$$

The tunable parameter in this power-law solution is $P_{x,B} - P_{x,Q}$. As $P_{x,B}$ may be measured from the landscape, the value of the fit should be able to be related directly to the exponent that describes the downstream increase in geomorphically effective stream discharge.

### 3.3 Fixed-width river

In order to generate an analytical solution for a fixed-width gravel-bed river, starting from Eq. (28), we assume that three key variables are constant (i.e., both steady and uniform): width ($b = B$, so $k_{b,B} = 1$), water discharge ($Q$), and grain size ($D$). As a result, $q = Q/b$ is also steady and uniform. This may be considered to be a short reach of an incised river

with no significant tributaries or a portion of an engineered canal for which discharge varies extremely gradually. Applying these assumptions, as well as assuming that $\tau_b^* \geq \tau_c^*$, produces the following nonlinear diffusion equation with a source/sink (uplift/subsidence) term:

$$\frac{\partial z}{\partial t} = \frac{21}{20} \frac{\phi g^{1/2} I}{1-\lambda_p} \left(\frac{\rho_s - \rho}{\rho}\right)^{1/2} (qD)^{3/5} \left(-\frac{\partial z}{\partial x}\right)^{-3/10} \tag{41}$$

$$\left(\frac{1}{\frac{\rho_s - \rho}{\rho}} \frac{0.345}{g^{3/10} \mathbb{S}^{7/10}} \left(-\frac{\partial z}{\partial x}\right)^{7/10} \frac{q^{3/5}}{D^{9/10}} - \tau_c^*\right)^{1/2} \frac{\partial^2 z}{\partial x^2} + U.$$

Solving this equation for the case in which any vertical motion is provided by uplift or subsidence (Eq. 38) is a general case of a steady-state long profile ($\partial z/\partial t = 0$) with no uplift or subsidence ($U = 0$). Applying this assumption defines a channel with a uniform slope, where $(x_0, z_0)$ is a point along the channel long profile,

$$z = z_0 - 4.57 g^{3/7} \mathbb{S} \left(\frac{\rho_s - \rho}{\rho}\right)^{10/7} \frac{\tau_c^{*10/7} D^{9/7}}{q^{6/7}} (x - x_0). \tag{42}$$

Slope adjusts to the driving force required to maintain a uniform bed-load sediment discharge. Increasing submerged specific gravity, $(\rho_s - \rho)/\rho$, and grain size, $D$, resist sediment motion by increasing the weight of the grains; therefore, the equilibrium fluvial transport slope is also increased. Increasing discharge per unit width ($q$), in contrast, decreases the equilibrium fluvial transport slope, as this provides more power to move the bed-material sediment.

## 4 Numerical solutions

To solve more general cases of Eqs. (20) and (28), we derive numerical solutions described in Appendix D. The solution to Eq. (20) (D3) is solved semi-implicitly by constructing equations with a diffusive component that can be solved directly in a tridiagonal matrix and a set of nonlinear terms that require Picard iteration. This solution method improves numerical stability and reduces computation times. Python code to solve for the shapes of river long profiles is available online at https://github.com/awickert/grlp (last access: 10 December 2018, Wickert, 2018). This library includes functions to analytically solve for the long profile shape as well (Eq. 40), and with the proper inputs, this can match the analytical solution (Fig. 2).

## 5 Discussion

### 5.1 Parameterizing stream-power-based sediment discharge

Whipple and Tucker (2002, Eq. 4) posited that sediment discharge should follow the power-law relationship

$$Q_c = K_t A^{m_t} S^{n_t}, \tag{43}$$

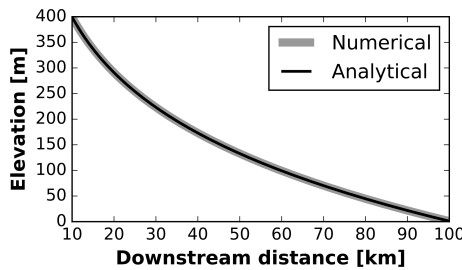

**Figure 2.** When $\mathrm{d}z/\mathrm{d}t = U$, the analytical solution for an equilibrium-width river (Eq. 40) matches the numerical solution for an equilibrium-width river (Eq. 36). Equation (36) is derived from the general equation for an equilibrium-width river, Eq. (20), to include power-law downstream relationships for valley width and water discharge (Eqs. 31–35). Here, the slope at the upstream boundary condition is $S_0 = 0.015$; this is set to produce an input bed-load sediment discharge of $Q_{s0} = 3.48 \times 10^{-4}$ m$^3$ s$^{-1}$. Water discharge, $Q = 1.43 \times 10^{-5} x^{49/40}$ m$^3$ s$^{-1}$; drainage area, $A = x^{7/4}$ m$^2$, and valley width, $B = 25x^{1/5}$ m.

where $Q_c$ is the bed-load sediment-transport capacity and is equal to $Q_s$ for transport-limited rivers, $K_t$ is a coefficient, $A$ is drainage area, and $m_t$ and $n_t$ are exponents. Howard and Kerby (1983) and Willgoose et al. (1991) present arguments for $m_t = n_t = 2$ for sand-bed rivers, and Whipple and Tucker (2002) posit that $n_t = 1$ for gravel-bed rivers.

The sediment-transport formulation that we present in Eq. (18), when combined with the discharge–drainage-area relationship of Eq. (31) and dropping references to directionality, can be rewritten in a way that is analogous to the above equation for $Q_c$:

$$Q_c = k_{Q_s} k_{A,Q} I A^{P_{A,Q}} S^{7/6}. \tag{44}$$

This relationship provides a value for $n_t$, based on our above derivation, which is grounded in sediment-transport experiments and morphodynamic theory (Meyer-Peter and Müller, 1948; Parker, 1978; Wong and Parker, 2006). It also provides a likely range of values for $m_t$ based on empirical studies that relate drainage basin area to geomorphically effective discharge. Furthermore, it defines a starting point towards quantifying the free parameter $K_t$: $k_{Q_s} = 0.041$ is known (Eq. 19), $I$ relates to the variability of the hydrograph, and $k_{A,Q}$ must relate to precipitation patterns across the drainage basin. Therefore, we focus on understanding the power-law drainage area–discharge scaling ($k_{A,Q}$ and $P_{A,Q}$), as solving this would constrain or define the remaining constants in Eq. (44) and allow us to relate slope and drainage area, easily measured from digital elevation models (DEMs), directly to gravel transport capacity.

The appropriate value of $P_{A,Q}$ depends on the flow of interest. For mean flow in a basin that experiences uniform precipitation, it is 1 (given catchment-wide water balance). For rarer flows, $P_{A,Q} < 1$. This is because smaller basins may be completely covered by a storm event, leading to a

catchment-wide response to a unit hydrograph, but larger basins may not have coherent storms across the whole basin, leading to attenuation of flood peaks and a decreased likelihood of a flood that is as large a ratio of the mean flow as in the small basin (Aron and Miller, 1978; Snow and Slingerland, 1987; Milly and Eagleson, 1988; Huang and Niemann, 2014). Aron and Miller (1978) found that, for annual flood peaks in $\sim 50$ streams in Pennsylvania and New Jersey (USA), $P_{A,Q} \approx 0.7$; such annual floods are generally also those that move gravel. Whipple and Tucker (1999) suggest values of 0.7–1.0 for bedrock erosion, and Sólyom and Tucker (2004) find that $1/2 \leq P_{A,Q} \leq 1$, which is in agreement with field data from Strahler (1964, p. 50). The lower limit from Sólyom and Tucker (2004) is for a single storm whose duration is much less than the time it takes for the water from the storm to travel through the basin. O'Connor and Costa (2004) used the entire U.S. Geological Survey gauging history (Slack and Landwehr, 1992) to compute that, on average, $P_{A,Q} = 0.57$ for 90th-percentile floods and $P_{A,Q} = 0.53$ for 99th-percentile floods.

We normalize $A$ to a characteristic footprint area of storms that occur across the catchment over the timescale of interest, $A_R$, and assume that $A \geq A_R$ for transport-limited gravel-bed rivers:

$$Q_c = k_{Q_s} I q_R A_R \left( \frac{A}{A_R} \right)^{P_{A,Q}} S^{7/6}. \tag{45}$$

This definition applies the power $P_{A,Q}$ to a dimensionless ratio, thereby ensuring that the coefficients can be framed in terms of rainfall. Here, we define a new coefficient that is the rainfall rate (i.e., flux) during a specific set of coincident rainfall events, $q_R$; $k_{A,Q} = q_R A_R^{1-P_{A,Q}}$. For simplicity, we do not consider inefficiencies in rainfall-to-discharge conversion, although factors could be added to an analogous expression to represent evapotranspiration and/or groundwater loss to other catchments.

From this relationship, we can assign values to the following parameters from Whipple and Tucker (2002):

$$K_t = k_{Q_s} I q_R A_R^{1-P_{A,Q}} \tag{46}$$
$$m_t = P_{A,Q} \tag{47}$$
$$n_t = 7/6. \tag{48}$$

For example, picking a characteristic storm footprint of $100 \, \text{km}^2$, $P_{A,Q} = 7/10$ (after Aron and Miller, 1978), and $q_R = 1 \, \text{cm} \, \text{h}^{-1}$, we find that $K_t \approx 2 \times 10^{-5} \, \text{m}^{2/7} \, \text{s}^{-1}$, $m_t = 7/10$, and $n_t = 7/6$. This provides a set of reasonable values for values that were left as free parameters in earlier derivations (Whipple and Tucker, 2002), demonstrates the relative importance of slope vs. drainage area in setting sediment discharge, and in Sect. 5.2 demonstrates how $m_t = P_{A,Q}$ and $n_t = 7/6$ set the concavity index of transport-limited gravel bed rivers.

## 5.2 Concave-up long profiles require weathering and/or downstream fining

Whipple and Tucker (2002) proposed that at steady state, sediment discharge should be proportional to uplift times contributing area by a constant, $0 \leq \beta \leq 1$. $\beta = 0$ indicates that all eroded material is removed as wash load or dissolved load. $\beta = 1$ indicates that all eroded material becomes bedload (i.e., gravel-sized) sediment.

We make the modification that contributing area must be raised to a power, $0 \leq P_\beta \leq 1$, that we term the "gravel persistence exponent". This describes the persistence of gravel-sized particles as they are weathered through hillslope processes (Attal et al., 2015; Sklar et al., 2017) and/or fine downstream to sizes that are smaller than gravel (Sternberg, 1875; Attal and Lavé, 2009; Dingle et al., 2017). If $P_\beta = 1$, every piece of eroded material on the landscape becomes gravel that reaches the stream. If $P_\beta = 0$, the amount of gravel reaching the stream is independent of drainage area. Intermediate values of $P_\beta$ indicate that some combination of hillslope weathering and downstream fining reduce the gravel supply to a nonzero fraction of the initially eroded material.

$$Q_s = \beta A^{P_\beta} U. \tag{49}$$

By assuming that channels are transporting sediment at capacity and that most transport-limited gravel-bed rivers should have gravel banks and therefore exist with an equilibrium width (following Eq. 7, i.e., $Q_s = Q_c$), we can set Eqs. (44) and (49) equal to one another, and rearrange the terms to create a slope–area relationship:

$$S = \left( \frac{\beta U}{k_{Q_s} k_{A,Q}} \right)^{6/7} A^{(6/7)(P_\beta - P_{A,Q})}. \tag{50}$$

For a river at steady state to have a concave long profile, meaning that channel slope decreases as drainage area increases (as is observed in nature), the exponent to which drainage area ($A$) is raised must be negative. This slope–area exponent, multiplied by $-1$, is defined as the concavity index, $\theta$, (Whipple and Tucker, 1999):

$$S = k_s A^{-\theta}. \tag{51}$$

Here, $k_s$ is the channel steepness index (Moglen and Bras, 1995; Sklar and Dietrich, 1998; Whipple, 2001). Together, steepness (coefficient) and concavity (exponent) define the power-law relationship for slope. Because slope is the $x$ derivative of elevation, this also implies that the channel long profile should be described by a power law, which is consistent with the analytical solution (Sect. 3.2).

In the case of Eq. (50), $\theta = -(6/7)(P_\beta - P_{A,Q})$. If $P_\beta = 1$, as assumed by Whipple and Tucker (2002, Eq. 7b), and $0.5 \leq P_{A,Q} \leq 1.0$, as prior work has demonstrated (Aron and Miller, 1978; Snow and Slingerland, 1987; Whipple and Tucker, 1999; O'Connor and Costa, 2004), then the exponent to which $A$ is raised would become positive. Such a

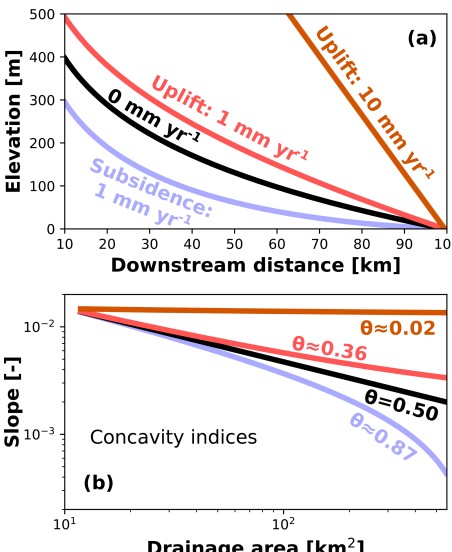

**Figure 3.** Steady-state numerical model outputs with steady uplift (base-level fall), subsidence (base-level rise), or neither. These numerical solutions are formulated following Eq. (D3), which is a finite-difference discretization of the general equation for an equilibrium-width transport-limited gravel-bed river, Eq. (20). Power-law relationships describe downstream increases in water discharge ($Q$) and valley width ($B$), following Eqs. (31)–(36). **(a)** Long profiles. All channels are plotted such that they are pinned to the same downstream point. **(b)** Slope–area plots: concavities increase with increasing subsidence. Model input parameters other than uplift are the same as those given for the long profiles displayed in Fig. 2.

river would be required to have a constant-to-downstream-increasing slope in order to transport the sediment that it is supplied. This would result in a straight-to-convex steady-state long profile, which runs contrary to common observations of natural channels.

These assumptions produce a convex long profile because as drainage area increases, sediment supply increases more strongly than water discharge. A straightforward solution is to adjust $P_\beta$, which describes the attenuation rate of gravel-sized particles with increasing drainage area. As drainage area increases, so does the mean transport distance of a particle that reaches the corresponding point on the stream. As transport distance increases, so does the possibility of significant weathering on the hillslope or breakdown in the channel (Attal and Lavé, 2009; Attal et al., 2015; Sklar et al., 2017; Dingle et al., 2017). This combination of weathering and downstream fining can significantly reduce the amount of gravel-sized sediment supplied to a channel cross section as drainage area increases.

An approximate value for the gravel persistence exponent, $P_\beta$, can be calculated by noting that in most natural rivers $\theta \approx 0.45$ to $0.5$. Combining this with the observation that $0.5 \le P_{A,Q} \lessgtr 0.7$ leads to the result that $P_\beta \lessgtr 0.2$. This

low gravel persistence exponent implies rapid attenuation of gravel-sized sediment as drainage area increases: doubling of the drainage basin area would produce a $< 15\%$ increase in the volume of gravel-sized sediment supplied to a channel cross section. For breakdown of clasts within the fluvial system, this is qualitatively consistent with the work of Dingle et al. (2017), who observed that most gravel produced in the Himalaya is converted into sand within 100 km travel distance.

Figure 3, with long profiles calculated using Eq. (20), indicates that uplift can act to reduce the concavity in the downstream direction. The range of applicable solutions is bounded by practical limitations: uplift rates must be appropriate for the channels to remain transport-limited, and subsidence rates must be low enough that they do not overwhelm the sediment supply and cause internal drainage to develop. Uplift also impacts sediment supply by increasing the steepness of the hillslopes, which increases hillslope sediment-transport rates and hence decreases the time available for weathering and soil formation (Attal et al., 2015), resulting in increased hillslope gravel supply. As increasing rates of uplift (or base-level fall) force the channel long profile towards a constant slope (concavity $\theta \to 0$), Eq. (50) demonstrates that the gravel persistence exponent, $P_\beta$, increases until it equals the drainage-area–discharge exponent, $P_{A,Q}$.

The small value of $P_\beta$ significantly increases the critical drainage area for the transition from a detachment-limited channel to a transport-limited channel (Whipple and Tucker, 2002). This is because increasing drainage area does not increase sediment supply as rapidly as assumed by (Whipple and Tucker, 2002). Therefore, a relatively larger portion of the landscape may be assumed to be detachment-limited than previously thought.

### 5.3 Concave-up long profiles may require valley widening

Equation (39) for a steady-state river with neither uplift nor subsidence can be rewritten with $dz/dx$ replaced by $S$ and $P_{x,Q}$ replaced by its constituent components $P_{x,A}$ and $P_{A,Q}$:

$$\frac{7}{6}\frac{1}{S}\frac{dS}{dx} = \frac{P_{x,B} - P_{x,A}P_{A,Q}}{x}. \tag{52}$$

In order to solve this equation, we rely on the fact that at the upstream boundary condition, $x = x_0$ and $S = S_0$. Here, the slope is set to prescribe the input sediment discharge, $Q_{s_0}$, in a way that is independent of the water discharge (see Eq. 18). We solve Eq. (52) to obtain a slope–distance relationship,

$$S = S_0 \left(\frac{x}{x_0}\right)^{(6/7)(P_{x,B} - P_{x,A}P_{A,Q})} \tag{53}$$

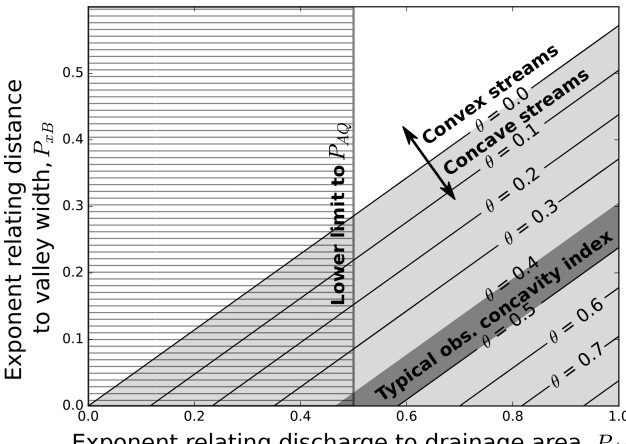

**Figure 4.** The slope–area concavity index defined in Eqs. (51) and (54) limits the range of possible powers for discharge–drainage area and width–distance relationships. The light gray field includes all concave long profile solutions, and the dark gray area indicates where the concavity index is in a commonly observed range for rivers in the field, between 0.4 and 0.5. The hatched area on the left is below the theoretical lower limit for the exponent that relates drainage area to water discharge, $P_{A,Q} = 0.5$, which exists in the limit where storm duration is much less than the time taken for water to pass through the catchment (Sólyom and Tucker, 2004). This example is given for an equilibrium-width river for which $\mathrm{d}z/\mathrm{d}t = U$, which corresponds to the analytically solvable case in Eqs. (39) and (40).

We then substitute drainage area, $A$, for $x$ based on an inversion of Eq. (32):

$$S = S_0 \frac{k_{x,A}^{(6/7)(P_{A,Q} - P_{x,B}/P_{x,A})}}{x_0^{(6/7)(P_{x,B} - P_{x,A} P_{A,Q})}} A^{(6/7)(P_{x,B}/P_{x,A} - P_{A,Q})}. \qquad (54)$$

Based on Eq. (54), the concavity index (Eq. 51) is $\theta = (6/7)(P_{A,Q} - P_{x,B}/P_{x,A})$, and the steepness index, $k_\mathrm{s}$ is equal to the terms forming the coefficient before the $A$ term. For a characteristic inverse Hack exponent ($P_{x,A} = 7/4$) and range of likely concavity index values, $0.4 \lesssim \theta \lesssim 0.5$, a tight bound exists on the possible values of $P_{A,Q}$ and $P_{x,B}$ (Fig. 4). These values span the range of observed (Aron and Miller, 1978; Howard and Kerby, 1983; Whipple and Tucker, 1999) and theoretical (Sólyom and Tucker, 2004) steady-state river concavity index values. Furthermore, this formulation demonstrates that a downstream-widening valley can be necessary to produce rivers of observed concavity index values for common values of $P_{A,Q}$. Insofar as valley widening can be recognized in the field, this observation can be used in areas of little to no uplift to connect geomorphic form directly to the area scaling relationship for a dominant river discharge (Fig. 4, dark gray diagonal region).

## 5.4 Signatures of change in the sediment-to-water supply ratio (climate) and/or base level (tectonics)

Transport-limited river-channel long profiles evolve in response to water and sediment inputs (e.g., Parker et al., 1998) and relative base-level change (Hilley and Strecker, 2005). Water and sediment inputs can occur both at the upstream boundary and throughout the catchment, and changes in relative base level occur at the downstream boundary. We find by applying the above derivation for an equilibrium-width gravel-bed river (Sect. 2.1) that such rivers adjust their steepness to the sediment-to-water input ratio (Sect. 5.4.1) and adjust their concavities to the rate of relative base-level change, such as that due to tectonic uplift or subsidence (Sect. 5.4.2). Increasing sediment supply dampens the magnitude of the concavity response by decreasing the relative contribution of the uplifting or subsiding valley floor to the sediment budget.

These distinct modes of response can help us to distinguish whether the river is responding primarily to changes in water and/or sediment supply, or to changes in base level. This may help to disentangle the effects of climate – often related to water and sediment supply (Tucker and Slingerland, 1997; Simpson and Castelltort, 2012) – and tectonics, which can change relative base level. However, tectonics may also affect sediment supply and grain size by modifying topographic relief (Attal et al., 2015; Sklar et al., 2017). Over longer timescales, tectonics may also increase or reduce water inputs by influencing orographic precipitation (e.g., Pingel et al., 2014). Other non-climatic factors – including human, biological, and (bio)geochemical activity – may also impact water and/or bed-material sediment delivery to rivers (e.g., Liébault and Piégay, 2001; James, 2013; Pelletier et al., 2015; Acosta et al., 2015; Sklar et al., 2016; Garcin et al., 2017). Furthermore, non-tectonic changes in base level, such as those caused by sea-level change, glacial isostatic adjustment, reservoir construction, dam removal, or climatically driven aggradation or incision of the main-stem river into which the study tributary flows, could contaminate a "tectonic" signal (Cantelli et al., 2004; Faulkner et al., 2016; Wickert, 2016). An important caveat to this is that many such natural base-level changes also change the horizontal position of the river outlet, and the overall river response is due to both horizontal and vertical changes in outlet position, even though we discuss only an idealized vertical base-level change here.

### 5.4.1 The sediment-to-water discharge ratio determines channel steepness

Uniform changes in the input sediment-to-water discharge ratio, in the absence of changes in the uplift rate (or equivalently, the rate of base-level change), determine the steepness index of a channel, but do not affect its concavity (Eqs. 51 and 54). As the input sediment-to-water discharge ratio increases, the channel steepens in order to transport the ad-

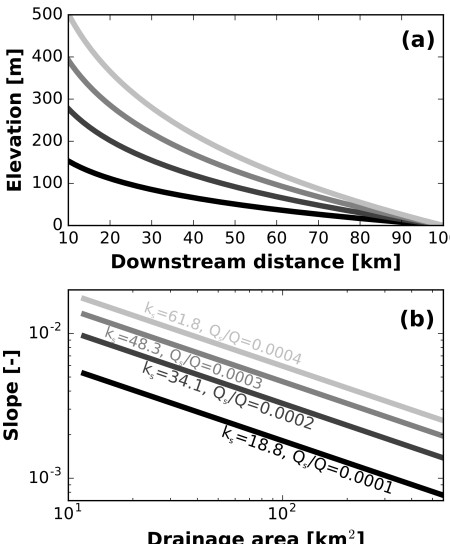

**Figure 5.** As the sediment-to-water discharge ratio increases, a steeper channel is required to mobilize the sediments, and as a result, the channel steepness index ($k_s$, Eq. 51) increases.

ditional sediment load out of the system at the rate that it is supplied (Figs. 5 and 6d). This increase in steepness and the associated aggradation is sourced at the upstream boundary and propagates downstream: all sediment within the computed long profile is transported at capacity, following Eq. (18); therefore, perturbations to the sediment supply must be sourced by either changing the input boundary condition, as we do here, or by adding sediment along the channel using the "uplift" source/sink term ($U$ in Eq. 20). Conversely, a decrease in the input sediment-to-water discharge ratio causes a downstream-propagating decrease in steepness (Fig. 6e). Changing the sediment-to-water discharge ratio requires adjusting the virtual slope at the upstream boundary ($S_0$). Thus, this steepening can also be viewed as the natural result of requiring the solution to the equation for the long profile to accommodate a steeper upstream gradient boundary condition.

### 5.4.2 Tectonic uplift and subsidence modulate river concavity

Changes in the rate of base-level rise or fall, including those caused by tectonic subsidence or uplift, modify the concavity but not the steepness of a transport-limited gravel-bed river long profile (Fig. 3), for a given input sediment-to-water discharge ratio. Rivers experiencing tectonic subsidence (base-level rise) will have more concave steady-state long profiles than those with no uplift or subsidence, and those experiencing uplift (base-level fall) will have straighter (less concave) long profiles (Fig. 3). This can be understood as follows: base-level rise "pushes" the bottom of the river profile

upwards, bending it, while base-level fall "pulls" the bottom of this curve downwards, straightening it.

The analytical solution (Eq. 40) provides a long profile in the absence of uplift or subsidence ($\mathrm{d}z/\mathrm{d}t = 0$). This solution follows the black line in Fig. 3 and can be a useful reference case against which to compare numerical solutions of long-profile shape. Numerical solutions, in contrast, demonstrate deviations in long-profile shape from this reference case in response to nonzero uplift and/or subsidence.

Even though uniform changes in the water-to-sediment discharge ratio cannot impact long-profile concavity on their own, they can (through volume balance) influence the degree to which rates of uplift (or subsidence) changes do. Uplift or subsidence add or remove material from the bed of the river, and changes in concavity are the river's response to redistribute sediment discharge to balance these local sources or sinks of sediment. If the sediment discharge of the river is large compared to the amount of material moved by uplift or subsidence, then only a small adjustment of concavity is necessary to balance this source (uplift) or sink (subsidence) and maintain steady-state topography. A river carrying very little sediment, however, will have to dramatically change its long-profile concavity in order to reach steady state. Therefore, the steady-state long-profile concavity (Fig. 8) results from a competition between tectonics and sediment discharge, in which a channel-concavity change is induced by a tectonic (or base-level) forcing, but is dampened by increasing sediment input.

In order to compare both sediment discharge and uplift using a dimensionless parameter, we define a characteristic alluvial response rate ($\mathbb{A}$) as a velocity scale to compare against the uplift rate. The alluvial response rate is the ratio of the incoming sediment discharge ($Q_{s_{in}}$) to the area of the valley floor, which in turn equals the mean valley width ($\overline{B}$) multiplied by the length of the study river segment ($L$). This is the maximum rate at which sediment- transport processes can cause the valley to aggrade, and also scales with the power of the river to export sediment and incise.

$$\mathbb{A} = \frac{Q_{s_{in}}}{L\overline{B}}. \tag{55}$$

Using SI units of length, $1/\mathbb{A}$ is the time that it takes the river to aggrade 1 m if no sediment is exported from the catchment.

We note that $Q_{s_{in}}$ is only equal to the incoming sediment discharge at the upstream boundary condition, $Q_{s_0}$, for the case in which $P_{x,Q} = P_{x,Q} = 0$, indicating that there are no tributaries. When implicitly considering tributary inputs of water and sediment, as we do for any nonzero $P_{x,A}$ and $P_{x,Q}$, the total sediment input can be calculated by imposing a steady-state assumption with no uplift, which requires that the total sediment output must equal the sediment input. This can be calculated using Eq. (18), with discharge at the downstream boundary known, and the slope at the downstream boundary calculated using Eq. (53).

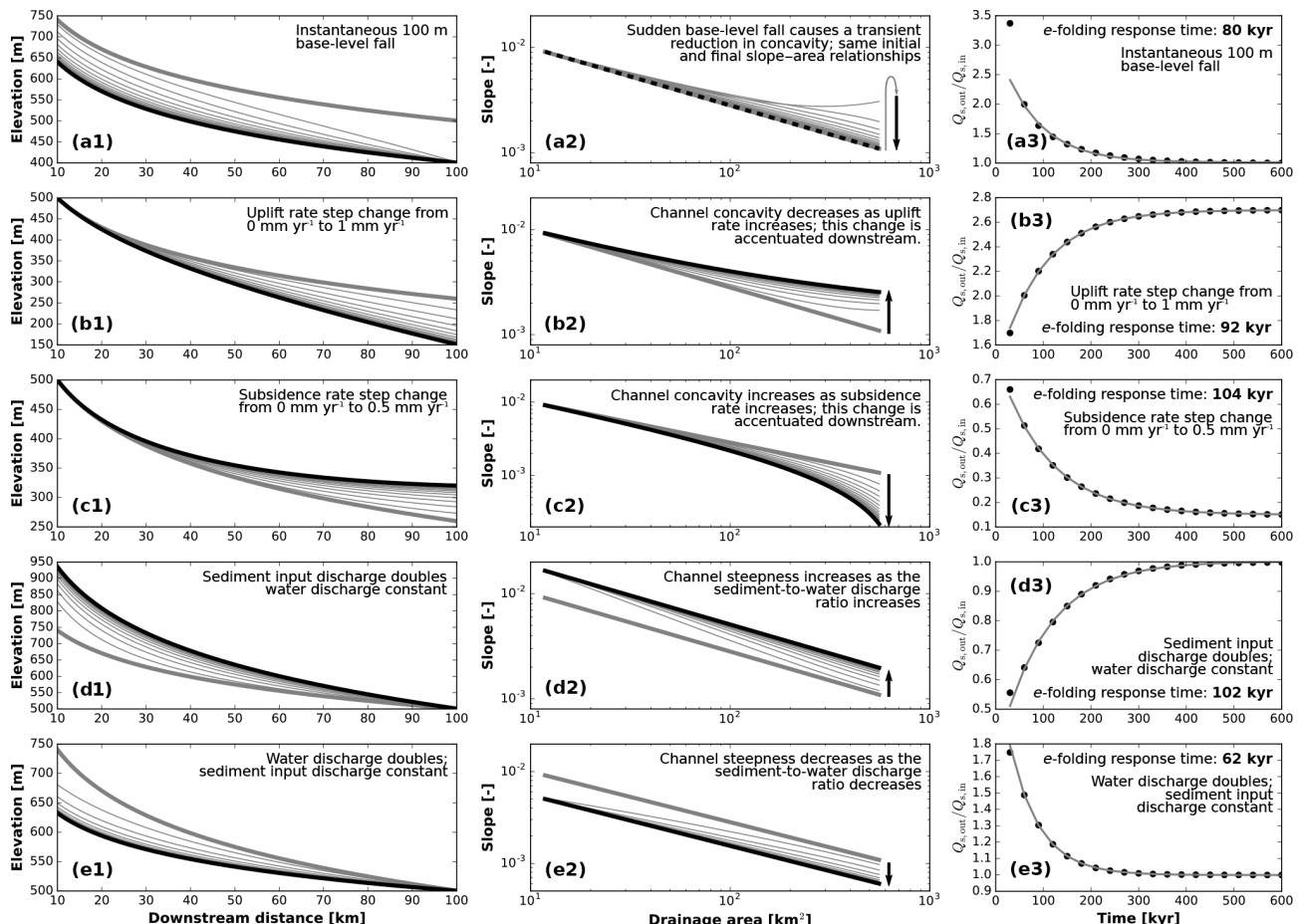

**Figure 6.** Transient long profiles from numerical model runs and their response times to external forcings. Each fine gray line in panel sets 1 and 2, corresponding to black dots on panel set 3, represents 30 000 years with an intermittency of $I = 1$ (i.e., constant geomorphically effective discharge conditions). Thick gray lines on panel sets 1 and 2 are the initial long profile; thick black lines are the final long profile. In the slope–area plots (panel set 2), the slope of the line is the negative of the concavity index and the $y$ intercept is the normalized steepness index (Whipple and Tucker, 1999). Transient response times (right) are quantified by the ratio of catchment-wide sediment input ($Q_{s,in}$) to output sediment discharge ($Q_{s,out}$). We quantify response times using exponential decay functions and $e$-folding timescales, but note that this does not describe the changes in sediment discharge immediately following the perturbation. All base-level changes are purely vertical, and can consequently represent steeply dipping faults or sea-level change across a steep coastline. Base model boundary conditions and parameters are as follows: $S_0 = 0.01$; $10\,\mathrm{km} \leq x \leq 100\,\mathrm{km}$; $Q = 1.43 \times 10^{-5} x^{49/40}\,\mathrm{m^3\,s^{-1}}$; $A = x^{7/4}\,\mathrm{m^2}$; and $B = 79.06 x^{1/10}\,\mathrm{m}$. **(a)** An instantaneous 100 m base-level fall causes a transient response and a sudden increase in sediment output but eventually produces the same channel long profile, albeit translated downward. **(b)** The onset of $1\,\mathrm{mm\,yr^{-1}}$ steady base-level fall (or tectonic uplift) reduces channel concavity; this allows the river to transport the additional bed-derived sediment as it incises and causes sediment output to rise as a result. **(c)** The onset of $0.5\,\mathrm{mm\,yr^{-1}}$ steady base-level rise (or subsidence) increases concavity due to increasing deposition rates that are required to fill the accommodation space created, and reduces sediment output accordingly. **(d)** Doubling the input sediment discharge ($Q_{s_0}$), facilitated by adjusting $S_0$ according to Eq. (18), increases channel steepness proportionally (Eq. 54); this increase in steepness propagates downstream and gradually leads to an increase in output sediment discharge. **(e)** Doubling the water discharge ($Q$) decreases channel steepness proportionally; this decrease in steepness propagates downstream more rapidly than that due to doubling sediment input because an increase in water discharge increases sediment-transport capacity; output sediment discharge increases until the slope decreases, returning sediment output to its initial rate.

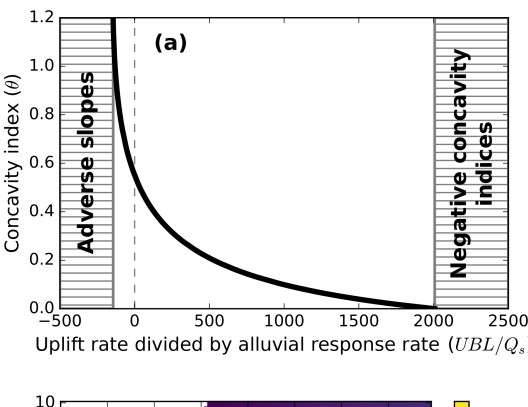

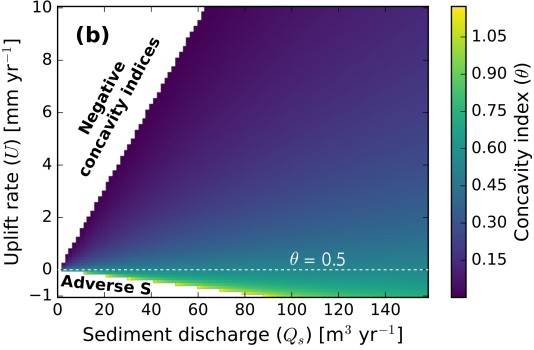

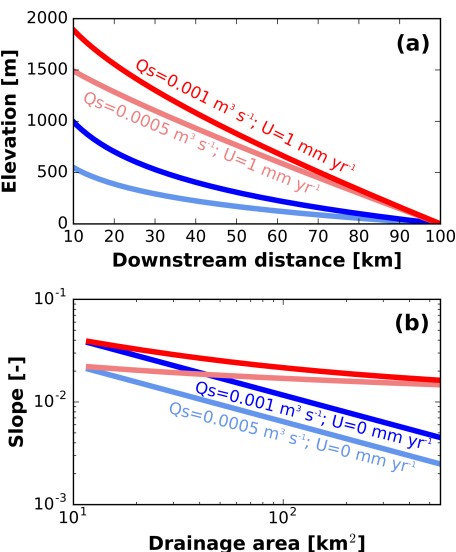

**Figure 8.** Covarying tectonic uplift (or base-level fall) and the input sediment-to-water supply ratio produces a range of channel long profiles **(a)** and steepness and concavity indices **(b)**. Changes in the tectonic uplift rate impact channel concavity indices, $\theta$, whereas changes in the water-to-sediment discharge ratio mainly impact channel steepness indices, $k_s$. A higher sediment supply dampens the effect of uplift on concavity. While these drivers and responses are distinct, tectonic uplift may increase sediment supply by steepening hillslopes; therefore, the variables controlling both the upstream (sediment supply) and downstream (base level) boundary conditions may change at the same time.

**Figure 7.** Concavity changes uplift (or subsidence) rates increase when compared to a characteristic alluvial response rate. **(a)** As in Fig. 3, increasing uplift rates decrease the channel concavity index. Here we vary channel width, sediment discharge, and uplift, and demonstrate how channel concavity change follows the ratio of uplift rate, the external driver, to the internal system response rate (Eq. 56). We disallow solutions that produce adverse slopes (these occur with high subsidence) or negative concavity indices (these occur with high uplift rates), as the former break the assumptions of our equations and the latter are not observed steady-state forms in nature. Changes in the valley-width exponent, $P_{x,B}$, change the shape of this curve by modifying the downstream distribution of valley widths and therefore altering the local alluvial response rates; all calculations for both panels were performed using $P_{x,B} = 1$. **(b)** For a single mean valley width (177 m), we compare the concavity index against the ratio of sediment discharge to uplift rate. It is important to note that with no uplift, concavity is constant at $\theta = 0.5$ regardless of sediment discharge.

Dividing the tectonic uplift (or subsidence) rate ($U$) by the alluvial response rate ($A$) provides a dimensionless number that defines the relative importance of sediment discharge vs. tectonics in determining the concavity of transport-limited gravel-bed rivers:

$$\frac{U}{\mathbb{A}} = \frac{LBU}{Q_s}. \tag{56}$$

As this ratio becomes more positive, concavity decreases; as it becomes more negative, concavity increases. Uplift (or subsidence) rate determines the existence and sign of the concavity change, whereas the ratio of uplift rate to the alluvial response rate determines the magnitude by which con-

cavity deviates from a reference value for a river that experiences no uplift; in Fig. 7, this reference value is 0.5.

Rivers also exhibit a transient response to changes in base level at a rate that is proportional to the alluvial response rate, $\mathbb{A}$ (Eq. 55). A single sudden change in base level generates a diffusive wave of incision (Fig. 6a1) or aggradation. This wave propagates upstream until the channel achieves the same slope and concavity as it did prior to the incision or aggradation event (Fig. 6a2), just at a different absolute elevation. A change in the rate of base-level change over time (through, for example, a change in tectonic uplift or subsidence rate) propagates upstream and changes the concavity of the river (Fig. 6b, c). We characterize the timescale of this response in terms of the ratio of the input vs. output sediment flux. When this ratio is less than unity, the river valley is storing sediment, and when it is greater than unity, it is releasing sediment. This change in sediment storage produces a disequilibrium change in the long-profile shape. Following the initial change, an exponential decay function can describe the approach to a new equilibrium state. This behavior allows us to define an *e*-folding response time that approximates the time required for a river system to respond to a perturbation (Fig. 6a3–e3).

### 5.4.3 Feedbacks between sediment supply and tectonics

In the above section, we have separated the effects of tectonics and climate as concavity and steepness responses, respectively. Our concavity changes derived from theory and their causes are generally consistent with the broad range of concavities and causes thereof synthesized by Whipple (2004, p. 161), albeit for bedrock rivers. However, such observations do not preclude a potential feedback by which increasing tectonic uplift rates may also increase gravel-sized sediment supply to the channel. In other words, the simplified approach of "climate = water-to-sediment supply, tectonics = base level" may be over-simplified.

Section 5.2 indicates that as uplift rates increase, the landscape surrounding the channel system steepens and erodes (Roering et al., 1999). Our above solutions for changes in tectonic uplift rates (Fig. 6b and b3) require only that the channel excavates the additional sediment from the bed of its valley. This excavation does not include additional sediment from the surrounding hillslopes, and steeper landscapes (often resulting from tectonic uplift) may be expected to produce a larger fraction of coarse material through landsliding and a shorter residence for weathering in the shallow subsurface (Attal et al., 2015; Carretier et al., 2015; Schildgen et al., 2016; Sklar et al., 2017). Changing gravel supply can dramatically alter river long profiles (Savi et al., 2016); therefore, an increase in the tectonic uplift rate may lead to both an increase in channel steepness that is unrelated to climate and a dampened decrease in concavity due to the increase in incoming bed-load sediment discharge that increases the alluvial response rate, $\mathbb{A}$ (Fig. 8). This tight channel–hillslope linkage challenges the paradigm that channel incision rates control hillslope morphology and motivates future work into models of landscape evolution that track and conserve sediment (Shobe et al., 2016; Sklar et al., 2017).

### 5.5 Concavity and downstream fining required for $b \propto Q^{1/2}$

It has long been recognized that river channel width scales with discharge to the 1/2 power,

$$b \propto Q^{1/2}. \tag{57}$$

This observation has been confirmed by a century of field studies (Lacey, 1930; Leopold and Maddock, 1953; Hey and Thorne, 1986; Singh, 2003). It has also been the subject of theoretical approaches to determine the static shape of a river channel (Savenije, 2003; Millar, 2005). Here we derive a physically based reason for this observation for an equilibrium-width gravel-bed river.

Equation (16) relates the width of an equilibrium-width gravel-bed river to discharge, slope, and grain size. Starting with Eq. (16), a slope–area relationship of $S = k_s A^{-\theta}$ (Eq. 51), and the discharge–drainage-area relationship from

Eq. (31), one can write that

$$b = k_b k_s^{7/6} k_{A,Q}^{(7/6)\theta/P_{A,Q}} \frac{Q^{1-(7/6)\theta/P_{A,Q}}}{D^{3/2}}. \tag{58}$$

This equation demonstrates that channel width is controlled by water discharge, channel concavity (through the concavity index), and downstream fining. Assuming a tight bound on channel concavity, as is generally assumed and has been observed in bedrock channels in the field (Duvall, 2004), although not universally (Whipple, 2004), two main drivers of channel width remain: water discharge and downstream fining of gravel-sized sediment (Fig. 9). Increasing water discharge can cause the channel to widen by requiring more space for the water to flow. Decreasing bed-material grain size reduces the critical Shields stress for the initiation of motion, and in order for an equilibrium-width channel to maintain a constant ratio of applied to critical Shields stress, the channel slope must become gentler and/or the channel itself must become wider. Due to the aforementioned tight bounds on the concavity index, the rate at which the channel slope decreases is also fixed, and any additional channel response to downstream fining must occur through channel widening.

Combining Eqs. (57) and (58) and simplifying the result produces a solution for a power that relates grain size to discharge, $P_{D,Q}$. This demonstrates how grain size must vary downstream in order to maintain the observed channel-width–discharge relationship:

$$D \propto Q^{(3-7\theta/P_{A,Q})/9}; \tag{59}$$

therefore,

$$P_{D,Q} = \frac{3}{9} - \frac{7}{9} \frac{\theta}{P_{A,Q}}. \tag{60}$$

The range of physically permissible values for the exponent that relates drainage area to discharge, $P_{A,Q}$, is 0.5–1.0 (Costa and O'Connor, 1995; Sólyom and Tucker, 2004). Combining this range of values with a typical concavity index of $\theta = 0.5$ produces bounds on the exponent in Eq. (59) of $-4/9 \le P_{D,Q} \le -1/18$: all plausible solutions require downstream fining to occur in order to reproduce the observed channel-width–discharge relationship (Lacey, 1930).

Using standard values of $\theta = 0.5$ and $P_{A,Q} = 0.7$, which is representative for a 1-year flood (Aron and Miller, 1978), one finds that $b \propto Q^{1/6} D^{-3/2}$. In this case, in order to recover the empirical $b \propto Q^{0.5}$ relationship, $D$ must be proportional to $Q^{-2/9}$. Testing this prediction against downstream fining data, requires that we convert discharge ($Q$) to distance downstream ($x$). While Sternberg (1875) provides reasoning to expect an exponential decay of grain size due to abrasion with distance downstream from a source area, this may be approximated by a power-law function, and downstream fining may also result from selective deposition (Whittaker et al., 2011). Multiplying $P_{A,Q} = 0.7$ by the inverse Hack exponent $P_{x,A} = 7/4$ (Eq. 34) produces the multiplier to con-

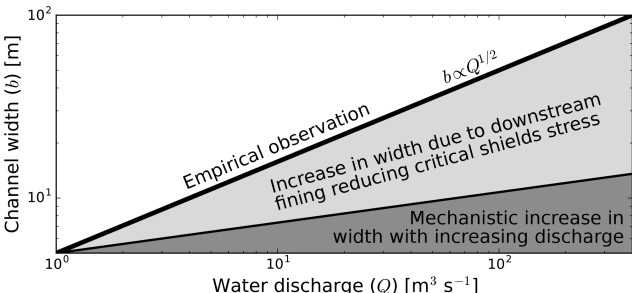

**Figure 9.** In an equilibrium-width self-formed gravel-bed river channel, the common field observation that river channel width is proportional to the square root of water discharge may be explained by a combination of the direct impact of river discharge on channel width and by downstream fining of bed-material sediment.

vert the grain-size–discharge relationship to a grain-size–downstream-distance relationship: $D \propto x^{0.27}$. We have not performed a rigorous analysis of this result, but data from Gomez et al. (2001) from the braided Waipaoa River in New Zealand are broadly consistent with this exponent.

## 6   Conclusions

We have produced equations to describe the long-profile evolution of transport-limited gravel-bed rivers by combining the Exner equation for conservation of volume, the Wong and Parker (2006) modification of the Meyer-Peter and Müller (1948) formulation for gravel transport, a Manning-style flow resistance equation (Parker, 1991), the normal-flow approximation for basal shear stress, the channel-width closure of Parker (1978), and the continuity equation. The key equation of this paper is Eq. (20), which captures the dynamics of a gravel-bed river whose bed shear stress is a multiple of the critical shear stress for initiation of motion; such systems are ubiquitous in nature (Phillips and Jerolmack, 2016; Pfeiffer et al., 2017). Furthermore, bedrock rivers can behave as transport-limited systems (Johnson et al., 2009), extending the applicability of our approach. Transport-limited gravel-bed river long profiles evolve more rapidly when they are steeper and/or experience a greater water discharge, and more slowly when their valleys are wider, as this requires that they fill more space. We solve Eq. (20) analytically for the special case in which $\mathrm{d}z/\mathrm{d}t = U$ – that is, that the river neither incises nor aggrades and does not respond to tectonic or base-level forcings. Both this solution and numerical solutions of steady-state rivers with constant uplift (or subsidence) rates have a power-law form, meaning that a power law can be appropriately fitted to transport-limited river long profiles.

Our derivation brings to light several significant relationships that may aid further efforts to understand river long profiles: (1) the sediment-transport formula for an equilibrium-width ($\tau_{\mathrm{b}}^*/\tau_{\mathrm{c}}^* = $ constant) gravel-bed river has the stream-power form proposed by Whipple and Tucker (2002). We quantify the values of its coefficient and exponents. The slope exponent is 7/6, and the other exponents relate to the scaling between the drainage area and the geomorphically effective discharge. (2) Gravel supply to rivers scales with uplift rate times contributing drainage area to a power that is less than 1, significantly modifying the implicit assumption of Whipple and Tucker (2002) that a uniform fraction of the sediment that is generated by rock uplift must be transported as bed load; this moves the expected position of the transition between detachment- and transport-limited long-profile evolution farther downstream. (3) Maintaining the observed slope–area scaling often requires that valleys widen downstream. (4) The water-to-sediment discharge ratio affects channel steepness, while the rate of base-level change affects channel concavity. This separation may allow the impacts of climate and tectonics to be separately inferred from channel long profiles, but increases in the uplift rate are often accompanied by increases in gravel-sized sediment supply via erosional processes (e.g., landsliding) associated with increasing landscape relief. Therefore, tectonic uplift can drive changes in long-profile shapes by inducing both base-level fall (reducing concavity) and an increase in sediment supply (increasing steepness). (5) The long-observed relationship that channel width increases as the square root of discharge (Lacey, 1930; Leopold and Maddock, 1953) can be explained through a combination of the direct effect of increasing discharge on equilibrium channel geometry and by downstream fining, which decreases the critical shear stress required to mobilize gravel on the channel bed and banks.

In this paper, we have derived a physics-based expression for the long-profile evolution of transport-limited gravel bed rivers, whose parameters are determined by theory, experimentation, and field work. We hope that this approach to understanding gravel-bed rivers provides forward momentum towards a more formal treatment of sediment transport and fluvial morphodynamics in river long-profile analysis and landscape evolution. Furthermore, by combining our derivation with other observations, we predict relationships among valley morphology, coarse-sediment production and evolution, and the power-law scaling between drainage area and geomorphically effective floods. While rivers are complex, we hope that these connections with broader pieces of the geomorphic puzzle can provide a path to build a better theory of fluvial system change and landscape evolution.

**Code availability.** The GitHub repository at https://github.com/awickert/grlp (last access: 10 December 2018, Wickert, 2018) contains the "grlp" Python module, which holds functions for both the analytical and numerical solutions presented here, as well as example implementation code to evolve transport-limited gravel-bed river long profiles.

## Appendix A: Notation

| | | |
|---|---|---|
| $\alpha$ | Angle of water surface and river bed with respect to horizontal | degrees or radians |
| $\beta$ | Gravel production coefficient | $\mathrm{m}^{3-2P_\beta}\,\mathrm{s}^{-1}$ |
| $\epsilon$ | Excess bed shear stress at bank-full flow ($\approx 0.2$) | – |
| $\gamma$ | Angle between down-channel flow direction and down-valley direction | degrees or radians |
| $\lambda_\mathrm{p}$ | Porosity ($\approx 0.65$) | – |
| $\lambda_\mathrm{r}$ | Roughness length scale (for flow resistance) | m |
| $\phi$ | Sediment-transport rate coefficient ($= 3.97$) | – |
| $\rho$ | Water density ($= 1000$) | $\mathrm{kg\,m}^{-3}$ |
| $\rho_\mathrm{s}$ | Sediment density ($= 2650$) | $\mathrm{kg\,m}^{-3}$ |
| $\tau_\mathrm{b}$ | Bed shear stress | Pa |
| $\tau_\mathrm{b}^*$ | Dimensionless bed shear stress (i.e., Shields stress) | – |
| $\tau_\mathrm{c}^*$ | Dimensionless critical shear stress (i.e., critical Shields stress) | – |
| $\theta$ | Channel concavity index (slope–area space) | – |
| $A$ | Drainage area | m2 |
| $\mathbb{A}$ | Alluvial response rate | $\mathrm{m\,s}^{-1}$ |
| $A_\mathrm{R}$ | Rainstorm footprint area | $\mathrm{m\,s}^{-1}$ |
| $b$ | Channel width | m |
| $b_x$ | Distance across valley cross section that is occupied by channel | m |
| $B$ | Valley width | m |
| $C_z$ | Chézy coefficient (for flow velocity) | – |
| $D$ | Grain size | |
| $g$ | Acceleration due to gravity ($= 9.807$) | $\mathrm{m\,s}^{-2}$ |
| $h$ | Flow depth | m |
| $I$ | Intermittency: fraction of time at geomorphically effective discharge | – |
| $k_{A,Q}$ | Coefficient to relate drainage area to water discharge | $\mathrm{m}^{3-2P_{A,Q}}\,\mathrm{s}^{-1}$ |
| $k_b$ | Threshold river width coefficient ($\approx 2.61$) | – |
| $k_{b,B}$ | Ratio between channel and valley width | – |
| $k_{q_\mathrm{s}}$ | Specific sediment-discharge coefficient ($\approx 0.0157$) | – |
| $k_{Q_\mathrm{s}}$ | Sediment-discharge coefficient ($= k_{q_\mathrm{s}} k_b \approx 0.041$) | – |
| $k_\mathrm{s}$ | Channel steepness index (slope–area coefficient) | $\mathrm{m}^{2\theta}$ |
| $k_{x,A}$ | Coefficient to relate distance downstream to drainage area | $\mathrm{m}^{P_{x,A}-1}$ |
| $k_{x,B}$ | Coefficient to relate distance downstream to valley width | $\mathrm{m}^{1-P_{x,B}}$ |
| $k_{x,Q}$ | Coefficient to relate distance downstream to water discharge | $\mathrm{m}^{3-P_{x,Q}}\,\mathrm{s}^{-1}$ |
| $K_t$ | Sediment-discharge capacity power-law coefficient ($= k_{Q_\mathrm{s}} q_\mathrm{R} A_\mathrm{R}^{1-P_{A,Q}}$) | $\mathrm{m}^{3-2m_t}$ |
| $L$ | River segment length | m |
| $m_t$ | Drainage area to sediment-discharge capacity exponent ($= P_{A,Q}$) | $\mathrm{m}^3\,\mathrm{s}^{-1}$ |
| $n_t$ | Slope to sediment-discharge capacity exponent ($= 7/6$) | $\mathrm{m}^3\,\mathrm{s}^{-1}$ |
| $P_\beta$ | Gravel persistence (resistance to weathering/fining) exponent | – |
| $P_{A,Q}$ | Power to relate drainage area to water discharge | – |
| $P_{D,Q}$ | Power to relate bed-material grain size to water discharge | – |
| $P_{x,A}$ | Power to relate distance downstream to drainage area (Hack exponent) | – |
| $P_{x,B}$ | Power to relate distance downstream to valley width | – |
| $P_{x,Q}$ | Power to relate distance downstream to water discharge | – |
| $q$ | Water discharge per unit channel width ($= \overline{u}h$) | $\mathrm{m}^2\,\mathrm{s}^{-1}$ |
| $q_0$ | Down-channel discharge per unit width (dummy variable: $= q$ or $q_\mathrm{s}$) | $\mathrm{m}^2\,\mathrm{s}^{-1}$ |
| $q_\mathrm{R}$ | Rainfall flux | $\mathrm{m\,s}^{-1}$ |
| $q_\mathrm{s}$ | Sediment discharge per unit channel width | $\mathrm{m}^3\,\mathrm{s}^{-1}$ |
| $q_{\mathrm{s},\hat{x}}$ | Down-valley sediment discharge per unit width | $\mathrm{m}^2\,\mathrm{s}^{-1}$ |
| $Q$ | Water discharge ($= qb$) | $\mathrm{m}^3\,\mathrm{s}^{-1}$ |
| $Q_\mathrm{c}$ | Sediment-discharge capacity ($= Q_\mathrm{s}$ if not supply-limited) | $\mathrm{m}^3\,\mathrm{s}^{-1}$ |
| $Q_\mathrm{s}$ | Sediment discharge (down-channel) ($= q_\mathrm{s}b$) | $\mathrm{m}^3\,\mathrm{s}^{-1}$ |

| | | |
|---|---|---|
| $Q_{s_{in}}$ | Combined sediment input from all tributaries | $m^3 \, s^{-1}$ |
| $Q_{s_0}$ | Upstream boundary condition sediment input | $m^3 \, s^{-1}$ |
| $Q_{s,\hat{x}}$ | Down-valley sediment discharge ($= Q_s$) | $m^3 \, s^{-1}$ |
| $S$ | Slope of water surface and river bed ($= \tan\alpha$) | – |
| $\mathbb{S}$ | Sinuosity (river length/valley length) | – |
| $S_0$ | Upstream valley slope boundary condition; sets sediment input | – |
| $t$ | Time | s |
| $\overline{u}$ | Mean flow velocity | $m \, s^{-1}$ |
| $U$ | Uplift (or subsidence) rate | $m \, s^{-1}$ |
| $x$ | Down-valley distance | m |
| $x_0$ | First (or only) known downstream distance for analytical solution | m |
| $x_1$ | Second known downstream distance for analytical solution | m |
| $\hat{x}$ | Down-valley unit vector | – |
| $z$ | Valley floor elevation | m |
| $z_0$ | Bed elevation at $x_0$ | m |
| $z_1$ | Bed elevation at $x_1$ | m |

## Appendix B: River valley width, channel sinuosity, and sediment balance

### B1 Width-resolving Exner equation

In the canonical Exner equation, the one-dimensional negative divergence in sediment flux is stated to be proportional to aggradation or incision, as given in Eq. (25). This one-dimensional form implies that the channel and valley width are the same (Fig. B1a) and that the valley walls are vertical and infinite. Such an approximation may not be bad in an artificial canal or for a rapidly incising river that consistently cuts a valley that is exactly one channel width. However, the former case is of less interest to geomorphologists and the latter case is more common in rapidly incising bedrock landscapes, where the rate of vertical incision greatly outpaces that of lateral erosion and valley widening.

In order to understand the evolution of a valley, one can first rewrite Eq. (25) through the definition of $q_s$ (Eq. 2) as follows:

$$\frac{\partial z}{\partial t} = -\frac{1}{1-\lambda_p}\frac{\partial (Q_{s,\hat{x}}/b)}{\partial x}. \tag{B1}$$

Here, $Q_{s,\hat{x}}$ is sediment discharge in the down-valley direction, which is always parallel to the down-channel direction when the valley and channel widths are the same. Explicit inclusion of channel width, $b$, provides a space to substitute the valley width, $B$, as the scale for the amount of material that must fill or be emptied from a cross section in order for the river to aggrade or incise:

$$\frac{\partial z}{\partial t} = -\frac{1}{1-\lambda_p}\frac{\partial (Q_{s,\hat{x}}/B)}{\partial x} \tag{B2}$$
$$= -\frac{1}{1-\lambda_p}\left(\frac{1}{B}\frac{\partial Q_{s,\hat{x}}}{\partial x} - \frac{Q_{s,\hat{x}}}{B^2}\frac{\partial B}{\partial x}\right).$$

This equation corresponds to Fig. B1b. For simplicity, this figure is drawn with a constant valley width, meaning that the second term on the right-hand side becomes 0 and the amount of time required to aggrade or incise is linearly dilated from that in Eq. (B1) (Fig. B1a) by the ratio $B/b$.

In order to solve this equation, we require a relationship that links the down-valley sediment discharge, $Q_{s,\hat{x}}$, to the downstream sediment discharge, $Q_s$. Both are obviously identical where the channel and valley are aligned (Fig. B1a, b). In the next section, we demonstrate that $Q_{s,\hat{x}} = Q_s$ for any arbitrary channel path that starts at the upstream end of a valley segment and ends at its downstream end. This is necessary for the final step to convert Eq. (B2) into Eq. (1).

### B2 The equivalence of downstream and down-valley discharge

Our main goal is to understand the evolution of valley networks, as channels perform the geomorphic work but valleys are the geomorphic units that evolve and constitute the

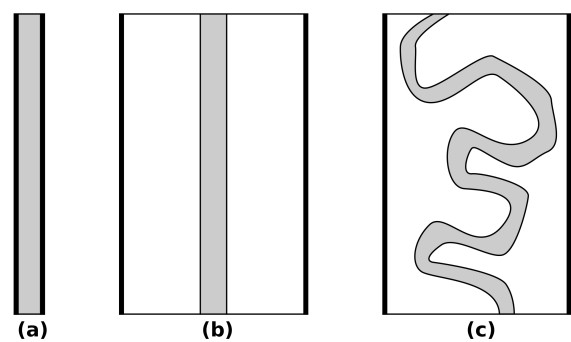

**Figure B1.** Our equations for sediment transport follow the river, but geomorphic evolution occurs along valley networks. In some cases (a), these are perfectly aligned and have the same width. However, if the river is aggrading and the valley walls are not vertical, and/or the river is eroding its valley walls at a rate that is comparable to or faster than its rate of vertical incision, (b) the river may carve a valley that is wider than its channel. (Here, the channel is pictured in the center, but it need not be positioned there.) In creating a wide valley, the river may move laterally, leading to (c) downstream and down-valley flow directions that are not aligned. While this depicts a single-thread channel, this same differentiation between channels and valleys is applicable to multi-thread channels.

broader landscape. In alluvial systems, valley geometries are not always identical to those of the channel networks that occupy them (Fig. B1), although they follow the same network-scale structure and connectivity. This possibility for non-alignment requires us to abandon the convenient choice of a channel-aligned coordinate system, typically used when solving for water or sediment discharge, and instead define our $x$ and $y$ coordinates to be down- and cross-valley, respectively (Fig. B2).

For an angle $\gamma$ between the down-valley and down-channel directions, a trade-off exists between the channel width occupying that cross section and the amount of sediment per unit down-valley width crossing it. Sediment discharge per unit width – which in map view can be represented by a vector – is reduced when the flow does not align directly with the valley.

$$q_{s,\hat{x}} = q_s\cos\gamma. \tag{B3}$$

For the same angle $\gamma$, the width of channel across the valley cross section increases:

$$b_{\hat{x}} = \frac{b}{\cos\gamma}. \tag{B4}$$

Through continuity, $Q_s = q_s b$ (Eq. 2). The $\cos\gamma$ terms in the above two equations cancel out, thus demonstrating that

$$Q_{s,\hat{x}} = q_s\cos\gamma\,\frac{b}{\cos\gamma} = q_s b = Q_s. \tag{B5}$$

If the channel crosses the valley multiple times (Fig. B2, top two and bottom plots), the total sediment discharge is simply

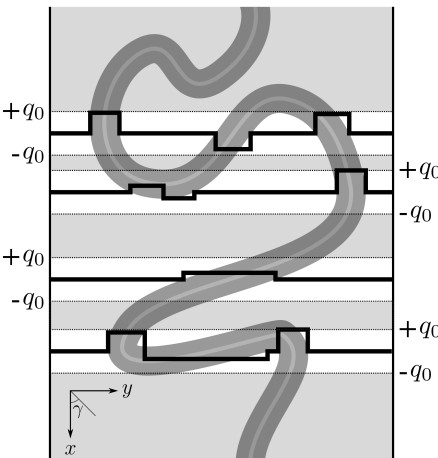

**Figure B2.** Down-valley discharge of water and sediment in each cross section is equal, and equals the down-channel discharge. Here, this is demonstrated geometrically, and results from discharge per unit width decreasing as the flow becomes more oblique to the valley cross section line, but the fraction of the total cross-valley line occupied by river channel increases in an inversely proportional manner. This remains true even for up-valley flow, which must be balanced with down-valley flow for water to be able to move from the upstream to the downstream ends of the valley. The direct down-valley discharge of water or sediment per unit width is given by $q_0$. In this schematic, discharge is given by a set of boxcar-function approximations (mean discharge across width) for the sake of simple illustration. While we use a single-thread river for simplicity, this same relationship holds for partitioning of flow and sediment transport across a multi-thread channel.

the sum of that in these cross sections, and this can be solved by summing across $y$, the cross-valley direction. Similarly, partitioning of flow and sediment across the branches of a multi-thread channel may be accounted for by summing $q_{s,\hat{x}}$ across $y$.

This geometric argument contains one mathematical caveat. Where a channel flows directly across the valley (i.e., in the $y$ direction), the solution for $Q_s$ is undefined. We address this problem by assigning a value of 0 to this undefined solution, based on our physical knowledge that flow that is neither up-valley nor down-valley will produce no discharge in the along-valley orientation.

By considering a continuity solution rather than a geometric one, it is possible to reason that our treatment of the above caveat is correct. In a system at steady state, each valley cross section must transmit downstream as much discharge as it is provided by the cross section immediately upstream. Therefore, discharge through every cross section must be equal, regardless of channel orientation. If this is the case, then a channel segment directed in line with the valley must transmit just as much water and sediment as a channel segment that is at an oblique angle to the valley axis. As a result, $Q_{s,\hat{x}} \equiv Q_s$.

Finally, while sediment (or any) discharge remains path-independent, the magnitude of this discharge is path-dependent. A more sinuous river course decreases channel slope (Eq. 5), and thus the driving stress for sediment transport. Therefore, the overall magnitude of both sediment discharge (Eqs. 18 and 27) and valley long-profile evolution (Eqs. 20 and 28) decrease with increasing sinuosity.

## Appendix C: Directionality

The full Meyer-Peter and Müller (1948) equation, with directionality included, is as follows:

$$q_s = \begin{cases} 0 & \text{if } |\tau_b^*| \leq \tau_c^* \\ -\text{sgn}\left(\frac{dz}{dx}\right)\phi\left(\frac{\rho_s - \rho}{\rho}\right)^{1/2} \\ g^{1/2}\left(|\tau_b^*| - \tau_c^*\right)^{3/2} D^{3/2} & \text{if } |\tau_b^*| > \tau_c^*. \end{cases} \tag{C1}$$

Here, sgn is the signum function (Eq. C2), and all other variables are described in Sect. 2.1 add Appendix A. The signum function, sgn, is defined as

$$\text{sgn}([\text{value}]) = \begin{cases} -1 & \text{if [value]} < 0, \\ 0 & \text{if [value]} = 0, \\ 1 & \text{if [value]} > 0. \end{cases} \tag{C2}$$

One additional difference between Eqs. (C1) and (3) is that Eq. (C1) incorporates the magnitude of $\tau_b^*$ as an additional result of relaxing the assumption that the river slopes downwards in the positive $x$ direction.

Including explicit directionality is not common when representing fluvial geomorphology mathematically. Slope, $S$, is typically used as a convenient shorthand for both its magnitude and direction. While we follow this convention in the main text to streamline the explanation, we include this explanation here because we have incorporated the signum function into the more general derivation, which accounts for directionality. This inclusion has allowed us to (1) relax the assumption that the downslope direction is known; (2) write the numerical model to self-consistently handle changes in flow direction; and (3) separate the sign and magnitude of the slope in equations that include a slope term raised to a power, thereby preventing a spurious imaginary part of the solution.

## Appendix D: Numerical solutions

### D1 Threshold-shear-stress river

Equation (20) has the form of a nonlinear advection–diffusion equation that can be rewritten for a numerical implementation as follows:

$$
\frac{\partial z}{\partial t} = \frac{k_{Q_s} I}{\mathbb{S}\left(1 - \lambda_p\right)} \left|\frac{dz}{dx}\right|^{1/6}
$$

(D1)

$$
\left[\frac{7}{6}\frac{Q}{B}\frac{\partial^2 z}{\partial x^2} + \frac{1}{B}\frac{\partial Q}{\partial x}\frac{\partial z}{\partial x} - \frac{Q}{B^2}\frac{\partial B}{\partial x}\frac{\partial z}{\partial x}\right] + U.
$$

For arbitrary $Q$–$x$ relationships and valley cross-sectional geometries ($B(z(x, t))$), and for solutions in which the valley geometry or discharge change with time ($B(z(x, t), t)$), a numerical solution becomes necessary. The above form of Eq. (20) can be solved semi-implicitly as

$$
z_{i,l} = 
$$

(D2)

$$
-\frac{\Delta t}{4(\Delta x)^2}\frac{k_{Q_s} I}{\mathbb{S}\left(1 - \lambda_p\right)}\left|\frac{z_{i+1,l^*} - z_{i-1,l^*}}{2\Delta x}\right|^{1/6}
$$

$$
\left[\frac{14}{3}\left(\frac{Q_{i,l} + Q_{i,l+1}}{B_{i,l}(z_{i,l}) + B_{i,l+1}(z_{i,l^*})}\right)\right.
$$

$$
\left(z_{i+1,l+1} - 2z_{i,l+1} + z_{i-1,l+1}\right)
$$

$$
+ \left(\frac{(Q_{i+1,l} + Q_{i+1,l+1}) - (Q_{i-1,l} + Q_{i-1,l+1})}{B_{i,l}(z_{i,l}) + B_{i,l+1}(z_{i,l^*})}\right)
$$

$$
\left(z_{i+1,l+1} - z_{i-1,l+1}\right) - \frac{\begin{matrix}(B_{i+1,l}(z_{i+1,l}) + B_{i+1,l+1}(z_{i+1,l^*})) \\ -(B_{i-1,l}(z_{i-1,l}) + B_{i-1,l+1}(z_{i-1,l^*}))\end{matrix}}{\left(B_{i,l}(z_{i,l}) + B_{i,l+1}(z_{i,l^*})\right)^2}
$$

$$
\left.\left(Q_{i,l} + Q_{i,l+1}\right)\left(z_{i+1,l+1} - z_{i-1,l+1}\right)\right] + z_{i,l+1} - U\Delta t.
$$

Here, $i$ is the $x$ index, $l$ is the $t$ index, and $\Delta x$ and $\Delta t$ are the spatial step and time step, respectively, assuming a uniform grid in space. The subscript $l^*$ of $z$ indicates that this term will be part of a Picard iteration: that is, it starts at $l$ and approaches $l + 1$ as multiple iterations of the solution provide sequentially better estimates of $z_{l+1}$.

Time-averaged values of $B$ and $Q$ are chosen to approximate conditions during the solution to the given time-step. Each of these can be simplified if $Q$ is known (as it typically is) or varies gradually in $t$ and/or $B$ varies gradually in both $z$ and $t$. Using notation that they are constant in time

$$
z_{i,l} = 
$$

(D3)

$$
-\frac{\Delta t}{4(\Delta x)^2}\frac{k_{Q_s} I}{\mathbb{S}\left(1 - \lambda_p\right)}\left|\frac{z_{i+1,l^*} - z_{i-1,l^*}}{2\Delta x}\right|^{1/6}
$$

$$
\left[\frac{14}{3}\left(\frac{Q_i}{B_i}\right)\left(z_{i+1,l+1} - 2z_{i,l+1} + z_{i-1,l+1}\right)\right.
$$

$$
+ \frac{Q_{i+1} - Q_{i-1}}{B_i}\left(z_{i+1,l+1} - z_{i-1,l+1}\right)
$$

$$
\left.- \frac{B_{i+1} - B_{i-1}}{B_i^2}Q_i\left(z_{i+1,l+1} - z_{i-1,l+1}\right)\right] + z_{i,l+1} - U\Delta t.
$$

This equation may be further simplified by moving one of the $(1/B_i)$ terms outside of the square brackets.

For an implicit solution, the terms inside the square brackets, plus $z_{i,l+1}$, constitute the stencil. The slope to the 1/6 power term outside of the stencil is a weak nonlinearity, and nonlinearities may also be introduced by changes in $B$ with $z$ and/or $t$. The uplift term modifies a Dirichlet boundary condition at the downstream end, and is analogous with base-level rise and/or fall.

A Neumann boundary condition of sediment-discharge input is used to set the slope at the upstream boundary using a "ghost-point" approach. This is solved for a defined $Q_s$ by rearranging Eq. (18) to define an upstream-boundary valley slope:

$$
S_0 = \mathrm{sgn}(q)\left(\frac{\mathbb{S}}{k_{Q_s} I}\frac{Q_s}{Q}\right)^{6/7}.
$$

(D4)

This equation demonstrates that slope increases with increasing sediment to water supply ratio, in agreement with the general principle of Lane's balance (Lane, 1955). For a domain that begins at 0,

$$
S_0 = -\left.\frac{dz}{dx}\right|_{x_0} \approx \frac{z_1 - z_{-1}}{2\Delta x}.
$$

(D5)

This equation can be rearranged to solve for the outside-domain elevation, $z_{-1}$ in terms of values inside the domain, and both the stencil and the right-hand-side column vector for the tridiagonal matrix solution can be updated accordingly.

### D2 Valley-width-controlled river

The general discretization of Eq. (28) for the long-profile evolution of a valley-width-confined transport-limited gravel-bed river is as follows:

$$
z_{i,l} = 
$$

(D6)

$$
- K_0\Delta t\left(K_1\left|\frac{z_{i+1,l^*} - z_{i-1,l^*}}{2\Delta x}\right|^{7/10}\frac{1}{D_i^{9/10}}\frac{Q_i^{3/5}}{b_i^{3/5}} - \tau_c^*\right)^{1/2}
$$

$$
D_i^{1/2}\left[\left|\frac{z_{i+1,l^*} - z_{i-1,l^*}}{2\Delta x}\right|^{-3/10}\left(\frac{3}{5}\frac{D_i^{1/10}}{Q_i^{2/5}b_i^{3/5}}\right)\right.
$$

$$
\left(\frac{Q_{i+1} - Q_{i-1}}{2\Delta x}\right)\left(\frac{z_{i+1,l+1} - z_{i-1,l+1}}{2\Delta x}\right)
$$

$$
- \frac{3}{5}\frac{Q_i^{3/5}D_i^{1/10}}{b_i^{8/5}}\left(\frac{b_{i+1} - b_{i-1}}{2\Delta x}\right)\left(\frac{z_{i+1,l+1} - z_{i-1,l+1}}{2\Delta x}\right)
$$

$$+ \frac{7}{10} \frac{Q_i^{3/5} D_i^{1/10}}{b_i^{3/5}} \left( \frac{z_{i+1,l+1} - 2z_{i,l+1} + z_{i-1,l+1}}{(\Delta x)^2} \right)$$

$$- \frac{9}{10} \frac{Q_i^{3/5}}{D_i^{9/10} b_i^{3/5}} \left( \frac{D_{i+1} - D_{i-1}}{2\Delta x} \right) \left( \frac{z_{i+1,l+1} - z_{i-1,l+1}}{2\Delta x} \right) \Bigg)$$

$$+ \left( K_1 \left| \frac{z_{i+1,l*} - z_{i-1,l*}}{2\Delta x} \right|^{7/10} \frac{1}{D_i^{9/10}} \frac{Q_i^{3/5}}{b_i^{3/5}} - \tau_c^* \right)^{1/2}$$

$$\left( \frac{D_{i+1} - D_{i-1}}{2\Delta x} \right) \Bigg] + z_{i,l+1} - U \Delta t.$$

Here, $K_0$ and $K_1$ are constants standing in for sets of sediment-transport-related terms in Eq. (28). This relationship is more nonlinear than that for the threshold-shear-stress river, above.

**Author contributions.** ADW wrote the paper, coupled the equations, and wrote the accompanying computer program. TFS motivated the paper through field studies; checked and revised the text, figures, and equations; supervised the project; and acquired funding for the research. Both authors contributed to the direction of the study and the discussion of its implications for fluvial geomorphology.

**Competing interests.** The authors declare that they have no conflict of interest.

**Acknowledgements.** This study was funded in large part by the Emmy-Noether-Programme of the Deutsche Forschungsgemeinschaft (DFG) grant no. SCHI 1241/1-1 awarded to Taylor F. Schildgen. Field and lab observations and discussions with Stefanie Tofelde, who revised an early version of the paper, helped to motivate the work. Conversations on channel concavity with Kelin Whipple and Greg Tucker stimulated initial thoughts on Sect. 5 of this paper, Chris Paola commented on an early form of the derivation, and Sam Holo pointed out an error in our treatment of sinuosity that appeared in the Discussions paper. Comments from Rebecca Hodge and one anonymous reviewer helped us to improve the paper.

Edited by: Tom Coulthard
Reviewed by: Rebecca Hodge and one anonymous referee

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
