# Peer review of "Long-Profile Evolution of Transport-Limited Gravel-Bed Rivers"

_Earth Surface Dynamics, 2018_

## Referee Comment (RC1) · Anonymous Referee #1 · 26 Jun 2018

This is a great paper, and I will be thinking about it, reading it, and citing it for a long time. It tackles some important questions in geomorphology that will be of very broad interest. I love the derivation of the width-discharge scaling relationship – wonderful! Also, I love that they tackled how to deal with the problem that power-law scaling in transport-limited 2D LEM models doesn't lead to concave channels – their solution seems spot on to me. Beyond the science, this paper is also very well written. I fully support publishing it, and it could be published in its current form.

Here I offer some general ideas and share where I got confused so that the authors can potentially add text here and there to answer some of my questions and minimize confusion. This is a very dense paper, so any reader is not going to get everything the first time. That's why I will be reading it for a long time to come. But that said, maybe

a few things can be clarified. I recognize that the authors know this work better than I do, so if they disagree with my suggestions it's fine. I don't think anything is technically wrong in the paper.

One of my big pet peeves, although I may be the only geomorphologist who thinks this, is calling slope the absolute value of the gradient. This can lead to sediment moving uphill. You use a few absolute values in your equations and then take care of the sediment moving uphill part by adding the signum function to your qs equation. But then eventually you get rid of it. I didn't understand why you did this. Looking at equation 3, (qs equation), I spent more time figuring out the sgn function, than if you had just changed your if statement. Why not just change the if statement that is already in the equation if to be if tb* <= tc*, then qs = zero? I went around and around on this, and I think you get to the same place by taking out the -sgn term and the absolute value on tau_b* by changing the if statement which to me is much simpler. Seeing the absolute value of bed shear stress was weird for me. And it took me a while to work out why that was needed when you also had sgn. It seemed overly complicated, but maybe I am missing something.

I know this seems like a micro-detail to bring up in my big comments, but when I have to go through 50+ equations, I don't want to be struggling on equation 3. And sgn is in other equations too. Why not just make the assumption that you are calculating in the downslope direction, as you are modeling 1D profiles anyway?

I also had a hard time going between general 2D network evolution and 1D channel evolution. I think the dz/dt equations should be general, and apply to a network, right? But the analytical solutions, which make the assumption that the channel is conveying sediment through, and not eroding or depositing, is for only 1D? Further confusing me, I think, is that when you bring up the Whipple and Tucker 2002, you are making an assumption that Qs is increasing downstream, so different from the earlier analytical solution. The W&T slope-area relationship – equation 51, and the 1D channel-only-conveying-what-is-sent-in-from-upstream slope-area relationship – equation 55, look

similar, but the idea that in one, only sediment is coming in from the top of the profile, and in the other there is a network producing sediment – seems like it should make a huge difference. I wonder if by adding the area exponent on the steady-state Qs relationship (eq. 50) but with such a low exponent – e.g. 0.2 – it basically shows that inputs of sediment from tributaries are less important for channel profile form than we thought? This is really hard for me to wrap my head around. I'm not sure I have fully appreciated this or if I am following. But possibly it is worth more discussion, or I missed the links in this study of your paper.

I know that you relaxed the assumption of only upstream sediment supply a bit in section 5.4.3, however this is very brief. I'm not sure what the answer is, and I hate to tell you to lengthen the paper, however, I did get confused moving among the different assumptions.

Details:

Page 2, Line 5: "However, such a ... " - Perfectly stated. I was hooked.

EQ 1 : Many people (I think) might be used to thinking of this equation only in terms of the first term in parentheses, and not the second one. Is it worth explaining each term?

P 8, First paragraph : Very nicely written and explained.

Eq 17 : It's not intuitive that channel width increases with slope. Is the explanation for that coming up? I immediately thought of Finnegan, Roe, Montgomery & Hallet, Geology, 2005. Might need some discussion.

P 10, L 1 : It finally dawned on me, should sediment discharge really be termed sediment transport capacity? Isn't qs what the channel can transport, but it will only transport if that sediment is available?

P 12, L 23 : I'm not sure you should cite Whipple and Tucker here. I don't think they show any data on the discharge-area relationship or Hack's law, they just use them.

P 14, L 12 : Do you mean constant? Or uniform? I think uniform as you say "a short reach ... with no significant tributaries"

Eq 44 : In the context of W&T 2002, I think what you have derived above is Qc, not Qs. It might be worth stating that. Maybe some people will get confused. Even in the appendix you call Qs sediment discharge, and Qc sediment discharge capacity. But it seems to me your equation 3, which your Qs equations comes from, is really a capacity, and in a sediment starved system this would not be the sediment discharge. Help me out please.

P 16, L 12 : I know one can't cite everything, but I particularly like the study by Huang and Niemann, 2014 to show this point. GSA Reviews in Engineering Geology, 2014 Simulating the impacts of small convective storms and channel transmission losses on gully evolution

P 17, L 6 – 10 : Is this discussion of P_beta actually beta? If P_beta equals zero, I think that means that the sediment flux is the same everywhere, or beta * U, and not that all material weathers on the hillslopes. Similarly, if P_beta = 1, but beta = 0, then no materialÂă reaches the stream as gravel. I'm confused.

Eq 50: Maybe also state bounds on Beta. Maybe obvious, maybe not?

P 18, Last paragraph : I'm a little lost in this paragraph... Are you talking about spatial variation in rock uplift? I don't think so, but "reduce the concavity in the downstream direction and "increases the fraction of the eroded landscape that acts to produce gravel" imply spatial variations to me. I think you mean that where rock uplift rate is lower, residence time is probably higher, so less gravel makes it to the channel. I guess this was also shown in Sklar et al, 2017?

Figure 6 : In this figure, is sediment input at the headwaters and sediment output at the outlet? I'm confused as to why these values assymptote to one when the sediment input varies or the base-level drops out for a period, but in the case of an uplift

rate change, it evolves to a new steady state in which sediment flux is increasing or decreasing downstream. Is this something related to Qs_in is fixed?

P 23, L 2,3 : This confuses me. Isn't P_beta in the concavity, and if that changes, wouldn't that change the input sediment-to-water discharge ratio?

P 24, L 12, 13 : I'm confused. I thought the first sentence of this section said that the sediment-to-water discharge ratio does impact long-profile concavity.

P 25 : "ÂăAs this ratio becomes more positive, concavity decreases; as it becomes more negative, the concavity decreases." Should the last word be increases?

P 28, L 18: Is this supposed to be P_DQ, not P_BQ?

P 28, L 21 : Am I following this correctly? I think you are saying that P_DQ needs to be -2/9, but that is outside the range that you predicted in the previous paragraph. If I'm right about that, then does that have implications for theta values and/or exponent values in the width-discharge relationship? If I'm wrong about this, I think I've gotten a bit lost.

P 29, L 26 : Not to be a pain in the ass, but this isn't strictly true. They had a coefficient on sediment production, but they didn't have the area-to-a-power scaling.

---

## Referee Comment (RC2) · R. Hodge (Referee) · 24 Aug 2018

This paper presents a new approach to predicting the long-profile evolution of gravel-bed rivers. The authors couple a range of governing equations for different components of these channels, resulting in a predictive relationship for channel development. From analysis of this relationship the authors are able to explain a number of observations including width-discharge scaling, and channel response to changes in sediment input, discharge and external forcing. I think that this paper will be of interest to a broad audience, and support its publication.

Overall I found the paper to be clearly written, although like the other reviewer I also found that there were places where I needed some more explanation. There are nec-

essarily a lot of equations in the paper and I have taken the authors' word that where they have combined equations etc. that this has been carried out correctly. The list of definitions at the end is useful, but there are places in the paper where it would be helpful to remind the reader what various parameters are.

One question that I had at a number of points is what the impact of size-selective entrainment would be on the model results. The model uses a single grain size, and the grain size is found to have to decease downstream in order to produce realistic concavity values. This decrease is implied to be caused by grain abrasion. However, we know that grain size also decreases downstream because of size-selective entrainment (e.g. Hoey and Ferguson. 1994). If you attributed the decrease in grain size to entrainment processes instead, would this have any impact on the rest of the model formulation? For example, abrasion should only be a function of transport distance, whereas the extent of size-selective entrainment will depend on the rate of sediment deposition.

The paper often refers to concavity and steepness, and it would be useful to state explicitly the relationship between the two. I assumed that changing the profile concavity would also change the slope of the profile (by different amounts at different locations), so I wasn't sure how the two could be seen as being separate from each other.

One general comment about the discussion is that in some places the figures are more extensively referred to by a section that is later on that the section that they are presented in. It might be worth double checking that all figures are in the most appropriate section and/or whether any sections could be combined.

Comments by page/line:

1/17: topographic relief of rivers or mountains?

2/28: Suggest replacing 'modifies' with 'defines'.

2/29: So how is your approach different to/an improvement on Blom et al?

3/25: Isn't the high excess shear stress also necessary to enable the river to erode

the bedrock bed as well as transport all the sediment? Also, is this something that you should come back to later on when assessing your model results from scenarios with an increase in uplift rates, as it suggests that your model assumptions might not apply in those conditions?

4/10: I got a bit confused by this material about the valley, probably because I would tend to think of long-profiles models as just considering the channel bed. It makes sense that if you want to raise the channel bed you also need to raise the elevation of all the material in the valley, otherwise the river will just occupy the lowest parts of the valley. However, you could state this more explicitly. In line 5/8 I wasn't sure if you were referring to the channel or the valley. I also wasn't sure where the terms in the brackets on the RHS of eq.1 had come from.

5/8: I think that this sentence about sinuosity will be clearer if you clarify the earlier material, but check; it took me a few reads to clarify what you meant. Also, sinuosity is introduced as a term here, but doesn't seem to feature in any of the later analysis. Is the impact of sinuosity on channel form (or the other way around?) something that you could look at in future work?

6/4: I agree with reviewer 1 that the use of the signum function is not intuitive. I can see why it might be useful to relax the assumptions, but I don't think that it is necessary is any of your analysis?

7/26 and 8/10: Could re-emphasise here that you are considering the excess shear stress and depth at the channel forming discharge.

8/28: I wasn't sure what I was meant to take away from that sentence.

11/9: It took me a couple of reads to get this comment about valleys not having vertical walls.

14/4: Rephrase

14/6: This is one of the points where I was trying to remember what the various P

parameters were. You have defined them in a sensible way, but it might help early on just to spell out your definition (e.g. that in all cases P_xy is the power that relates x to y).

16/2: It's not clear what studies you are referring to here.

16/31: What should I take from this example?

17/30: If weathering can reduce the amount of gravel, presumably it also alters the size?

18/5: Couldn't the location of the gravel-sand transition be as much a size-selective transport phenomenon rather than an abrasion-of-gravel phenomenon? I thought that Dingle et al actually supported your idea by the observation that the amount of gravel leaving a basin didn't seem to depend on basin size, and therefore most of the gravel from the basin was abraded to sand before leaving the mountain front.

18/8: I wanted a bit more explanation as to how Fig 3 was produced. It wasn't clear to me whether the increase in P_beta was falling out of the equations, or was something that you were altering.

19/5: If this model is for transport-limited conditions, can it be applied to these upper parts of the network?

20/8: I think that Fig 4 shows that valley widening is likely, but there is still a solution when P_xB is zero.

20/25: Amplitude of what?

21/fig 5: Explain in the caption which of the thick black/grey lines is the start/end. Why is there a dashed line in a2?

22/fig 6: I initially read the caption as being the ratio of sediment input to water output discharge, so clarify this sentence. Also, why do b and c not get to a state where the input and output sediment fluxes are equal?

24/3: Another steepness/concavity confusion; looking at fig 3, different slopes seem to be associated with different concavities.

24/21: This first phrase was not clear to me.

24/22: Here and after equation 56 are the first explicit mention of tributaries. I think that their input is implied in many of the earlier relationships, so it might be useful to mention them when presenting the earlier sections.

24/25: Might be useful to state that this time is that taken to fill the valley floor to a depth of 1 m?

25/3: One of these decreases should be an increase.

26/19: I think that you have implied this point earlier, but this is the first time that it is spelt out. Move to earlier on?

27/fig 9: I needed a bit more explanation to understand how this figure supported the point made in the text.

29/14: Does whether bedrock rivers behave as transport limited depend on the timescales over which you are considering them? One of the main assumptions about bedrock rivers is that over long timescales they are supply limited.

29/28: Is there any field evidence that identifies the location of the detachment- to transport-limited transition? How does it agree with your finding?

Reference: Hoey, T. B. and Ferguson, R. I.: Numerical-simulation of downstream fining by selective transport in gravel-bed rivers - model development and illustration, Water Resour. Res., 30(7), 2251–2260, 1994.

---

## Author Comment (AC1) · 22 Sep 2018

*Reviewers 1 (anonymous) and 2 (Hodge) offer many valuable thoughts to improve our manuscript. We thank them for their careful effort, which has helped us to make this dense manuscript more readable. In addition, both of the reviewers and Sam Holo have identified relatively minor but important mistakes, and in some cases corrected them. We are especially grateful for these notes.*

*Our responses to these comments are below. Quotes from the referees are in* Roman *font. Our responses are in italics. Quotes from our LaTeX source – typically modifications to the manuscript text – are in* `monospace` *font that is also italicized.*

**Responses to comments from Reviewer 1**

One of my big pet peeves, although I may be the only geomorphologist who thinks this, is calling slope the absolute value of the gradient. This can lead to sediment moving uphill. You use a few absolute values in your equations and then take care of the sediment moving uphill part by adding the signum function to your qs equation. But then eventually you get rid of it. I didn't understand equation 3, (qs equation), I spent more time figuring out the sgn function, than if you had just changed your if statement. Why not just change the if statement that is already in the equation if to be if tb* $\leq$ tc*, then qs = zero? I went around and around on this, and I think you get to the same place by taking out the -sgn term and the absolute value on tau_b* by changing the if statement which to me is much simpler. Seeing the absolute value of bed shear stress was weird for me. And it took me a while to work out why that was needed when you also had sgn. It seemed overly complicated, but maybe I am missing something.

I know this seems like a micro-detail to bring up in my big comments, but when I have to go through 50+ equations, I don't want to be struggling on equation 3. And sgn is in other equations too. Why not just make the assumption that you are calculating in the downslope direction, as you are modeling 1D profiles anyway?

*We fully agree that the any absolute value requires a signum function. The signum disappears in equation 21 because it is included in the $\partial z/\partial x$ term (which has a sign) in the first term in the brackets (and the term on the RHS outside the brackets). Furthermore, the absolute values on Q become unnecessary due to the fact that $(1/Q)\partial Q/\partial x$ can have only positive solutions for a real Q. This explanation aside, it appears that I can do a few things to help make this portion of the paper easier to follow!*

*The code for the model, as well as Equation 21, both include directionality. Therefore the key to solving this problem should be to shed the bulkiness (and hence confusion) caused by having these more general equations, while not losing their accuracy. The best way I can think of doing this is by moving the discussion with the signum function to an appendix. This will take some work, but I have gone through the text and begun to shorten/simplify the section on the derivation, while moving the essential information on directionality to an appendix. I maintain directionality in Equation 21 by referring to the appendix.*

I also had a hard time going between general 2D network evolution and 1D channel evolution. I think the dz/dt equations should be general, and apply to a network, right? But the analytical solutions, which make the assumption that the channel is conveying sediment through, and not eroding or depositing, is for only 1D? Further confusing me, I think, is that when you bring up the Whipple and Tucker 2002, you are making an assumption that Qs is increasing downstream, so different from the earlier analytical solution. The W&T slope-area relationship – equation 51, and the 1D channel-only- conveying-what-is-sent-in-from-upstream slope-area relationship – equation 55, look similar, but the idea that in one, only sediment is coming in from the top of the profile, and in the other there is a network producing sediment – seems like it should make a huge difference. I wonder if by adding the area exponent on the steady-state Qs relationship (eq. 50) but with such a low exponent – e.g. 0.2 – it basically shows that inputs of sediment from tributaries are less important for channel profile form than we thought? This is really hard for me to wrap my head around. I'm not sure I have fully appreciated this or if I am following. But possibly it is worth more discussion, or I missed the links in this study of your paper.

*The 1D vs. 2D question is a great point, and your confusion may be the result of my own mistake in describing the analytical solution. Since there are quite a few querstions / points here, I will have to break this response up in order to make sure that I address them all.*

*1D VS 2D: We parameterized a full network with a set of power-law relationships between discharge, drianage area, and downstream distance, without explicitly solving for a 2D network. These create a ficticious continuously-increasing discharge, whereas rivers typically experience jumps of increases in discharge at tributary junctions. You are correct that these equations could be generalized, and to do so would require internal boundaries between 1D profiles at tributary junctions. This is indeed an ongoing project!*

*ANALYTICAL SOLUTION: I (Wickert) made a mistake in describing this that I think was the source of all of this confusion. Because I wrote the power-law functions to relate discharge to downstream distance, and the equations assume transport and capacity, then there must be increasing sediment input with discharge downstream as well. Therefore, what I am instead indicating is that all of the hillslopes, both inside and outside the computational domain, are providing an eternal supply of sediments to the channel at transport capacity. This contradicts what I wrote:*

```
For such a no-uplift steady-state condition to persist over
geologic time requires a constant input of sediment from upstream
.
```

*Therefore, I have changed this text to:*

```
For such a no-uplift steady-state condition to persist over
geologic time requires a constant input of sediment from the
hillslopes. This may be reasonable for a river that reaches an
equilibrium long profile much more rapidly than the surrounding
landscape evolves and its relief changes. It is also useful as a
benchmark for numerical solutions (Figure \ref{fig:
AnalyticalNumerical}).
```

*WHIPPLE & TUCKER 2002: We are allowing sediment discharge to increase downstream, following water discharge, and removing my flub on the analytical solution description should (we hope) make this clearer. Both Equation 51 and Equation 55 include sediment from the full landscape. This is because sediment discharge in this transport-limited case is always directly proportional to water discharge, and water discharge in turn is proportional to drainage area.*

*TRIBUTARY SEDIMENTS AND $P_\beta$: The low exponent actually indicates that locally-derived sediments should be more important – its power (pun not intended!) to reduce sediment inputs increases as drainage area goes up. Therefore, it should imply that the tributary loads are important, and may replace sediments that are abraded, or, as Reviewer 2 (Hodge) points out, are preferentially deposited.*

I know that you relaxed the assumption of only upstream sediment supply a bit in section 5.4.3, however this is very brief. I'm not sure what the answer is, and I hate to tell you to lengthen the paper, however, I did get confused moving among the different assumptions.

*We hope that it is now clear that there was always locally-sourced sediment in a pseudo-2D sense; thank you for your above comments and for identifying my error.*

EQ 1 : Many people (I think) might be used to thinking of this equation only in terms of the first term in parentheses, and not the second one. Is it worth explaining each term?

*Yes, we agree that it is! Indeed, Exner and many others used just the 1D form of this equation, but it is significant to discuss how I expanded it for valley filling and/or emptying. We have added a new appendix to describe this. This appendix also describes the source of the sinuosity term, which we are correcting in the revision.*

Eq 17 : It's not intuitive that channel width increases with slope. Is the explanation for that coming up? I immediately thought of Finnegan, Roe, Montgomery & Hallet, Geology, 2005. Might need some discussion.

*The difference is that we are holding shear stress ratio constant (per Parker 1978) and Finnegan et al. are focusing on continuity; we have modified the text to describe this:*

```
This is the width created by a channel that has a constant ratio
of basal Shields stress to critical Shields stress (Equation \ref
{eq:ThresholdSelfFormedWidth} \citep{Parker1978}, and Equation \
ref{eq:b} predicts that for a given grain size and discharge, a
steeper channel will be wider in order to reduce depth and
therefore reduce applied shear stress. If the channel does not
```

*widen to maintain a constant excess shear stress ratio, \citep{*
*Finnegan2005} predict that it should narrow instead in response*
*to steepening because, holding bed roughness constant, steeper*
*flow is faster and therefore does not require as large a channel*
*width to transmit a given water discharge.*

P 10, L 1 : It finally dawned on me, should sediment discharge really be termed sediment transport capacity? Isn't qs what the channel can transport, but it will only transport if that sediment is available?

*These exist as separate entities in detachment-limited rivers, but are necessarily the same in transport-limited rivers. In order to make this clear early-on, I have added the following text to the description of the Meyer-Peter & Müller sediment transport relationship:*

*This formula is technically for sediment-transport capacity,*
*$Q_c$, per unit channel width, but in a transport-limited river,*
*sediment is always supplied at or above capacity such that $Q_s \equiv Q_c$.*

P 12, L 23 : I'm not sure you should cite Whipple and Tucker here. I don't think they show any data on the discharge-area relationship or Hack's law, they just use them.

*You are correct: I included it because it was one of my (Wickert's) personal introductions to these concepts; reference removed.*

P 14, L 12 : Do you mean constant? Or uniform? I think uniform as you say "a short reach ... with no significant tributaries"

*We mean both steady and uniform – uniform for the reason that you give here, and steady in order to find a time-invariant analytical solution. We have clarified this in the text.*

Eq 44 : In the context of W&T 2002, I think what you have derived above is Qc, not Qs. It might be worth stating that. Maybe some people will get confused. Even in the appendix you call Qs sediment discharge, and Qc sediment discharge capacity. But it seems to me your equation 3, which your Qs equations comes from, is really a capacity, and in a sediment starved system this would not be the sediment discharge. Help me out please.

*As noted above, transport-limited equations work only in non-sediment-starved systems, so the two are interchangeable... EXCEPT here, where we note the boundary case! Your comments have made me (Wickert) think that this may*

*not be clear to the community that works across the broader spectrum of channels. Therefore, we define $Q_c$ at the first place where we note $Q_s$, as noted above (MPM).*

P 16, L 12 : I know one can't cite everything, but I particularly like the study by Huang and Niemann, 2014 to show this point. GSA Reviews in Engineering Geology, 2014 Simulating the impacts of small convective storms and channel transmission losses on gully evolution

*This is fantastic – thanks! We had been looking for a more recent reference and had not found this one.*

P 17, L 6 – 10 : Is this discussion of P_beta actually beta? If P_beta equals zero, I think that means that the sediment flux is the same everywhere, or beta * U, and not that all material weathers on the hillslopes. Similarly, if P_beta = 1, but beta = 0, then no material reaches the stream as gravel. I'm confused.

*Thank you for catching this error. I (Wickert) stand by the $P_\beta = 1$ statement, because I state that "every piece of eroded material" (so this requires erosion). However, the $P_\beta = 0$ statement is indeed incorrect because it implies a drainage-area-independent sediment input, as you note here. I have corrected this, and also removed the sentence about gravel packing because on re-reading, I found it confusing and superfluous even though it seemed important to me at the original time of writing.*

Eq 50: Maybe also state bounds on Beta. Maybe obvious, maybe not?

*I added this information.*

P 18, Last paragraph : I'm a little lost in this paragraph... Are you talking about spatial variation in rock uplift? I don't think so, but "reduce the concavity in the downstream di- rection and "increases the fraction of the eroded landscape that acts to produce gravel" imply spatial variations to me. I think you mean that where rock uplift rate is lower, res- idence time is probably higher, so less gravel makes it to the channel. I guess this was also shown in Sklar et al, 2017?

*The reduction in concavity results from the fact that the valley floor becomes a local sediment source – and hence, the valley (and channel) must maintain a steeper slope in the channel's downstream reaches to balance the ever-increasing amount of local sediment input. (Downstream attenuation of this bed-derived sediment is not taken into account, and therefore this steepening may be a maximum estimate.) The statement about increasing the fraction of the eroded landscape is confusing because of our ambiguous use of "this", which is now replaced by a proper noun, "uplift", and additional explanation:*

`Uplift also impacts sediment supply by increasing the steepness`

*of the hillslopes, which increases hillslope sediment transport*
*rates and hence decreases the time available for weathering and*
*soil formation \citep{Attal2015}, resulting in increased*
*hillslope gravel supply.*

Figure 6 : In this figure, is sediment input at the headwaters and sediment output at the outlet? I'm confused as to why these values assymptote to one when the sedi- ment input varies or the base-level drops out for a period, but in the case of an uplift rate change, it evolves to a new steady state in which sediment flux is increasing or decreasing downstream. Is this something related to Qs_in is fixed?

*INLET VS. OUTLET: The input sediment is catchment-wide, and we have updated the text to reflect this:*

*Transient response and response times to external forcings as*
*quantified by the ratio of catchment-wide sediment input ($Q_\\*
*text{s,in}$) to output sediment discharge ($Q_\text{s,out}$).*

*DIFFERENTIAL RATIOS: The tectonics cases involve a local source (up-lift) or sink (subsidence) of sediment, whereas the river internally adjusts to changing sediment supply. We checked the caption and we are happy with our explanation of this; please let us know if you have an idea of how we can make this clearer.*

P 23, L 2,3 : This confuses me. Isn't P_beta in the concavity, and if that changes, wouldn't that change the input sediment-to-water discharge ratio?

*Good point – this is clarified to "uniform changes"*

P 24, L 12, 13 : I'm confused. I thought the first sentence of this section said that the sediment-to-water discharge ratio does impact long-profile concavity.

*We have updated some of the wording in this paragraph and the first para-graph in this section to make it clearer that this is for uniform changes: that is, I effectively set the upstream boundary-condition slope to set the incoming sediment supply to the system. This does not include local sources/sinks, which are in the same term as uplift/subsidence.*

P 25 : "As this ratio becomes more positive, concavity decreases; as it becomes more negative, the concavity decreases." Should the last word be increases?

*Yes!*

P 28, L 18: Is this supposed to be P_DQ, not P_BQ?

*Yes! Thank you.*

P 28, L 21 : Am I following this correctly? I think you are saying that P_DQ needs to be −2/9, but that is outside the range that you predicted in the previous paragraph. If I'm right about that, then does that have implications for theta values and/or exponent values in the width-discharge relationship? If I'm wrong about this, I think I've gotten a bit lost.

*This is in fact within the range that we predicted in the previous paragraph – −4/9 to −1/18. (Thank you for mentally correcting my typo!)*

P 29, L 26 : Not to be a pain in the ass, but this isn't strictly true. They had a coefficient on sediment production, but they didn't have the area-to-a-power scaling.

*Quite right! Changed "all" to "a uniform fraction of".*

**Responses to comments from Reviewer 2 (Hodge)**

The list of definitions at the end is useful, but there are places in the paper where it would be helpful to remind the reader what various parameters are.

*I (Wickert) have added a few more definitions, but I am not sure that I have them in the places that you would like them. I hope that by adding the appendices (see response to Reviewer 1) that the equations are now more streamlined and that this satisfies some of the spirit of this comment as well.*

One question that I had at a number of points is what the impact of size-selective entrainment would be on the model results. The model uses a single grain size, and the grain size is found to have to decease downstream in order to produce realistic concavity values. This decrease is implied to be caused by grain abrasion. However, we know that grain size also decreases downstream because of size-selective entrainment (e.g. Hoey and Ferguson. 1994). If you attributed the decrease in grain size to entrainment processes instead, would this have any impact on the rest of the model formulation? For example, abrasion should only be a function of transport distance, whereas the extent of size-selective entrainment will depend on the rate of sediment deposition.

*This is a great question, and the answer involves a combination of (1) correcting a misconception, and (2) considering the some scientific thought behind your suggestion. Towards (1), decreasing grain size does not alone impact concavity: it is decreasing grain size to the point that grains are carried in high suspension. The equations are insensitive to grain size so long as all of the grains are gravel. Towards (2), losing gravel would mean that there was an effective sediment sink; this could be lumped into the uplift/subsidence term and cause an increase in concavity as the river slope decreased to transport the smaller sediment supply at equilibrium. As noted by Hodge, this term would be nonuniform and a function of transport distance in the case of abrasion, whereas it could be related to subsidence rate for deposition (though size-selective deposition should matter less unless this affects the gravel–sand transition). I do not plan on covering (2) in this already-gigantic paper, but I do think that relationships between long-profile shape and grain-size evolution would be a good application for this approach.*

The paper often refers to concavity and steepness, and it would be useful to state explicitly the relationship between the two. I assumed that changing the profile concavity would also change the slope of the profile (by different amounts at different locations), so I wasn't sure how the two could be seen as being separate from each other.

*The current text that describes these two terms is:*

`In order for a river at steady state to have a concave long`
`profile, meaning that channel slope decreases as drainage area`
`increases (as is observed in nature), the exponent to which`

```
drainage area ($A$) is raised must be negative. This slope--area
exponent, multiplied by $-1$, is defined as the concavity index,
$\theta$, \citep{Whipple1999}:
\begin{equation}
S = k_s A^{-\theta}.
\label{eq:theta}
\end{equation}
Here, $k_s$ is the channel steepness index \citep{Moglen1995,
Sklar1998, Whipple2001}.
```

*To better explain the separation between these parameters, I have added two additional sentences at the end:*

```
Together, steepness (coefficient) and concavity (exponent) define
 the power-law relationship for slope. Because slope is the $x$-
derivative of elevation, this also implies that the channel long
profile should be described by a power law, which is consistent
with the analytical solution (Section \ref{s:threshold_analytical
}).
```

One general comment about the discussion is that in some places the figures are more extensively referred to by a section that is later on that the section that they are presented in. It might be worth double checking that all figures are in the most appropriate section and/or whether any sections could be combined.

*I (Wickert) looked through the text and found instances of this in regards to the figures of river transient response and their respective time scales. However, in these cases, they are referenced in three sections that are nonetheless neighboring. Therefore, I have moved these figures to the bottom of the section where they first appear, and as close as I feel to be reasonable to the next figure, which forms a maximum distance that I can move these in order to keep the figures in the same order that they are mentioned in the text. I hope that the typesetting of the two-column version will help with some of this presentation as well.*

1/17: topographic relief of rivers or mountains?

*Of mountains – I (Wickert) hadn't thought of topographic relief of rivers before. I changed one of the first sentences, using "topographic" in order to avoid using "mountain" twice; let me know if this is ambiguous and I will revisit it.*

```
Such rivers build and maintain topographic relief by carrying
gravel out of the mountains
```

2/28: Suggest replacing 'modifies' with 'defines'.
*Thank you! That is much better.*

2/29: So how is your approach different to/an improvement on Blom et al?

*We have added a clearer description of this:*

*In particular, we (1) consider evolution of the full river valley
, permitting analysis of time-scales longer than those of channel
 filling; (2) follow \citep{Parker1978} in allowing channel
widths to self-form as a function of excess channel-forming shear
 stress; and (3) define channel roughness as a function of flow
depth and grain size. (2) and (3) ultimately contribute to grain
size canceling out of the final equation, leading to a relatively
 simple and applicable equation for gravel-bed river long-profile
 evolution in response to changes in water supply, sediment
supply, and base level.*

3/25: Isn't the high excess shear stress also necessary to enable the river to erode the bedrock bed as well as transport all the sediment? Also, is this something that you should come back to later on when assessing your model results from scenarios with an increase in uplift rates, as it suggests that your model assumptions might not apply in those conditions?

*A high excess shear stress enables the river to erode bedrock iff bedrock is exposed. Here, Pfeiffer et al. look at reaches that have beds and banks of mobile material, though in such settings should, as you write, erode into bedrock as well. The point of these sentences is that these rivers need not have near-threshold shear stress, so I do not think I can write something about this here without diluting the paragraph. However, I obviously must agree with your general point that incising rivers will eventually connect with bedrock, and if they do and spend a comparable or greater amount of geomorphic work eroding the bedrock as compared to moving sediment, they become closer to the detachment-limited endmember. I have added the following text to describe*

*The range of applicable solutions is bounded by practical
limitations: uplift rates must be appropriate for the channels to
 remain transport-limited, and subsidence rates must be low
enough that they do not overwhelm the sediment supply and cause
internal drainage to develop.*

4/10: I got a bit confused by this material about the valley, probably because I would tend to think of long-profiles models as just considering the channel bed. It makes sense that if you want to raise the channel bed you also need to raise the elevation of all the material in the valley, otherwise the river will just occupy the lowest parts of the valley. However, you could state this more explicitly. In line 5/8 I wasn't sure if you were referring to the channel or the valley. I also wasn't sure where the terms in the brackets on the RHS of eq.1 had come from.

*In response to both yourself and Reviewer 1, I have added an appendix to address this. I think this to be an important issue because the standard paradigm in eroding landscapes to think just of the channel only works when vertical incision rates are much greater than lateral erosion rates, forcing the channel into a narrow space. Including the separation between the time-evolving river valley, which integrates past processes and is important for mass balance, and the river channel, which is the engine and driver, is key. This new appendix also includes a figure to describe this, and a correction to our handling of sinuosity. The only*

*thing that is not answered here is your confusion in line 5/8, because I cannot see the ambiguity – if this remains ambiguous, please let me know!*

5/8: I think that this sentence about sinuosity will be clearer if you clarify the earlier material, but check; it took me a few reads to clarify what you meant. Also, sinuosity is introduced as a term here, but doesn't seem to feature in any of the later analysis. Is the impact of sinuosity on channel form (or the other way around?) something that you could look at in future work?

*I have updated this section and also describe sinuosity in a new appendix. I do not consider this strongly here, but I do think that the co-evolution of sinuosity and valley width could be important, with the latter (I believe) being the more important variable.*

6/4: I agree with reviewer 1 that the use of the signum function is not intuitive. I can see why it might be useful to relax the assumptions, but I don't think that it is necessary is any of your analysis?

*I agree that this is not completely necessary, and is rather part of my (Wickert's) personality to write really complete general equations. Therefore, I have moved this discussion to an appendix as well. (See also the comments by Reviewer 1 and my replies; both of you were unified in your thoughts about this.*

7/26 and 8/10: Could re-emphasise here that you are considering the excess shear stress and depth at the channel forming discharge.

*Done and thank you; these are the kinds of points that are difficult to see when one is too close to the paper.*

8/28: I wasn't sure what I was meant to take away from that sentence.

*I added:*

`, thus making it an equally accurate and more mathematically convenient approach.`

11/9: It took me a couple of reads to get this comment about valleys not having vertical walls.

*I added:*

`Therefore, changes in valley elevation produce changes in valley width, even in absence of time-evolution of the valley geometry that then feeds back into the rate of long-profile evolution.`

14/4: Rephrase

*I can see how "Its solution is a law" wasn't clear. What a difference a word makes – it was meant to read "power law". I have updated the text to reflect this (and the later major typo) and to add a bit more explanation.*

`Its solution is a power law, solved using two known points along`
`the long profile -- ($x_0,z_0$) and ($x_1,z_1$). Practical`
`choices for these points are the upstream and downstream`
`boundaries of the river segment being studied.`

14/6: This is one of the points where I was trying to remember what the various P parameters were. You have defined them in a sensible way, but it might help early on just to spell out your definition (e.g. that in all cases P_xy is the power that relates x to y).

*Great idea – I have added the following text:*

`In order to write these in a consistent and intuitive way, all`
`power-law coefficients are designated $k$ and all exponents (``
`powers'') are designated $P$. Each pair is given an ordered pair`
`of subscripts that indicates first the variable that one is`
`converting from, and second the variable that one is converting`
`to.`

16/2: It's not clear what studies you are referring to here.
*Since the studies are listed, I am interpreting this to mean that it is unlcear why I am referring to them – please let me know if I have misunderstood. I have clarified the text to read:*

`based on our above derivation, which is grounded in sediment-`
`transport experiments and morphodynamics theory`

16/31: What should I take from this example?

*I have added the following sentence:*

`This provides a set of reasonable values for values that were`
`left as free parameters in earlier derivations \citep{Whipple2002`
`}, demonstrates the relative importance of slope vs. drainage`
`area in setting sediment discharge, and in Section \ref{s:`
`ConcavityRequirements} demonstrates how $m_t = P_{AQ}$ and $n_t =`
` 7/6$ set the concavity index of transport-limited gravel bed`
`rivers.`

17/30: If weathering can reduce the amount of gravel, presumably it also alters the size?

*Indeed it should, but this is not important here because the grain size cancels out. Towards what I perceive to be the spirit of your point, I can imagine a more general set of equations that include transport of multiple size classes,*

*considerations of hiding functions and equal mobility, etc., that may cause size-selective transport. Here, combining the width closure with sediment transport in a multi-D system would require some additional research that can connect hiding functions, channel-forming discharge, width closure, etc. in these situations.*

18/5: Couldn't the location of the gravel-sand transition be as much a size-selective transport phenomenon rather than an abrasion-of-gravel phenomenon? I thought that Dingle et al actually supported your idea by the observation that the amount of gravel leaving a basin didn't seem to depend on basin size, and therefore most of the gravel from the basin was abraded to sand before leaving the mountain front.

*Good point; we have rephrased this as follows to focus on abrasion:*

```
this is qualitatively consistent with the work of \citet{
Dingle2017}, who observe that most gravel produced in the
Himalaya is converted into sand within 100 km travel distance in
the Himalaya
```

18/8: I wanted a bit more explanation as to how Fig 3 was produced. It wasn't clear to me whether the increase in P_beta was falling out of the equations, or was something that you were altering.

*The caption describes that $P_\beta$ is not altered (and therefore falls out of the equations), but we have also altered the paragraph in the main text describing this as follows:*

```
Figure \ref{fig:UpliftSubsidence}, with long profiles calculated
using Equation \ref{eq:dzdt}, indicates that uplift can act to
reduce the concavity in the downstream direction.

As increasing rates of uplift (or base-level fall) force the
channel long profile towards a constant slope (concavity $\theta
\rightarrow 0$), Equation \ref{eq:SA_uplift_sed_supply}
demonstrates that the gravel persistence exponent, $P_\beta$,
increases until it equals the drainage-area-to-discharge exponent
, $P_{AQ}$.
```

19/5: If this model is for transport-limited conditions, can it be applied to these upper parts of the network?

*This relationship is appropriate here because I am comparing sediment supply to sediment-transport capacity, as has been done before to find the transition zone between transport- and detachment-limited systems.*

20/8: We think that Fig 4 shows that valley widening is likely, but there is still a solution when P_xB is zero.

*True; updated the "requires valley widening" in the section heading to "may require valley widening"*

20/25: Amplitude of what?

*Updated text to:*

`The magnitude of the concavity response is controlled in part by sediment supply.`

21/fig 5: Explain in the caption which of the thick black/grey lines is the start/end. Why is there a dashed line in a2?

*We reorganized one sentence in the caption and added this:*

`Thick gray lines are the initial long profile; thick black lines are the final long profile.`

22/fig 6: I initially read the caption as being the ratio of sediment input to water output discharge, so clarify this sentence. Also, why do b and c not get to a state where the input and output sediment fluxes are equal?

*Sentence clarified following this suggestion and that of reviewer 1. b and c being unequal is because of the uplift/subsidence being a local source/sink of sediment. As I noted to reviewer 1, I had thought this to be clear in the caption, and on re-reading it cannot think of how to make it clearer. If you still think it to be unclear and have ideas during a re-review, please let me know.*

24/3: Another steepness/concavity confusion; looking at fig 3, different slopes seem to be associated with different concavities.

*Hopefully the additional introduction of the steepness index, as being separate from the slope itself, will have clarified this.*

24/21: This first phrase was not clear to me.

*We have clarified this by revising the paragraph:*

`In order to compare both sediment discharge and uplift using a dimensionless parameter, we define a characteristic alluvial response rate ($\mathbb{A}$) as a velocity scale to compare against uplift rate. The alluvial response rate is the ratio of the incoming sediment discharge ($Q_{s_{\text{in}}}$) to the area of the valley floor, which in turn equals the mean valley width ($\bar{B}$) multiplied by the length of the study river segment ($L$). This is the maximum rate at which sediment transport processes can cause the valley to aggrade, and also scales with the power of the river to export sediment and incise.`

24/22: Here and after equation 56 are the first explicit mention of tributaries. I think that their input is implied in many of the earlier relationships, so it might be useful to mention them when presenting the earlier sections.

*Yes, and this is a good point! I have added the following text to the section introducing the power-law relationships:*

`These equations are continuum idealizations of a river with a`
`tributary network. Real rivers experience discrete jumps in water`
` discharge at tributary junctions. The smooth curves of water`
`discharge vs. downvalley distance produced by these relationships`
`, on the other hand, are beneficial for building intuition.`

*In the present section, we further clarify by removing the first offhand mention of tributaries and later revising the text to read:*

`We note that $Q_{s_{\text{in}}}$ is only equal to the incoming`
`sediment discharge at the upstream boundary condition, $Q_{s_0}$,`
` for the case in which $P_{xQ}=P_{xQ}=0$, indicating that there`
`are no tributaries.`

24/25: Might be useful to state that this time is that taken to fill the valley floor to a depth of 1 m?

*Sure: we can compute $1/\mathbb{A}$ to obtain the time scale that you suggest, and include this as follows:*

`Using SI units of length, $1/\mathbb{A}$ is the time that it`
`takes the river to aggrade 1 meter if no sediment is exported`
`from the catchment.`

25/3: One of these decreases should be an increase.

*Thank you; fixed per Reviewer 1.*

26/19: I think that you have implied this point earlier, but this is the first time that it is spelt out. Move to earlier on?

*We have edited some above wording to highlight the common idea that water-to-sediment discharge ratio is related to climate, whereas base-level changes are related to tectonics.*

27/fig 9: I needed a bit more explanation to understand how this figure supported the point made in the text.

*We have updated its caption to be much more descriptive:*

*Covarying tectonic uplift (or base-level fall) and input sediment
-to-water supply ratio produces a range of channel long profiles
(a) and steepness and concavity indices (b). Changes in tectonic
uplift rate impact channel concavity indices, $\theta$, whereas
changes in water-to-sediment discharge ratio mainly impact
channel steepness indices, $k_s$. A higher sediment supply
dampens the effect of uplift on concavity. While these drivers
and responses are distinct, tectonic uplift may increase sediment
 supply by steepening hillslopes, and therefore cause the
variables controlling both the upstream (sediment supply) and
downstream (base level) boundary conditions may change at the
same time.*

29/14: Does whether bedrock rivers behave as transport limited depend on
the timescales over which you are considering them? One of the main assumptions about bedrock rivers is that over long timescales they are supply limited.

*As I understand it, the question is more about the partitioning of geomorphic work: whether more goes to eroding the bedrock or more goes to moving the sediment. However, I do see transport- and detachment-limited rivers as endmembers, and the edge cases are hard to define. I have left the text as-is, still citing the same paper, because I don't think that I can reasonably tackle the broader question of how to appropriately apply such approximations in the conclusions section.*

29/28: Is there any field evidence that identifies the location of the detachment-to-transport-limited transition? How does it agree with your finding?

*In a short search, I (Wickert) have not been able to find the necessary literature data to test this. However, one could conceivably combine long profiles of transport-limited rivers with the positions of this transition to test this theory, with the caveat that the observed bedrock–alluvial transition may only approximate the transport-limited-to-detachment-limited transition point.*

**Response to comments by S. Holo**

Sam Holo brought up concerns about a possible mistake in how we included the sinuosity term. He was correct that there was a mistake, and we have now corrected this throughout the paper. Because including sinuosity is not immediately straightforward, We have included a description of how to do so in the same new appendix where we describe the effort to create a valley-resolving Exner equation. Fortunately, we ran all calculations with a sinuosity of 1, meaning that these all remain valid. We thank Sam for taking the time to read the paper and his help in improving it.